# Stringent sustainability regulations for global supply chains are supported across middle-income democracies

E. Keith Smith [1,3] ✉, Dennis Kolcava[2,3] & Thomas Bernauer [1]

Expanded international trade and globalised production networks are increasing the environmental and social impacts in middle-income countries (GNI per capita $1,136-$13,845). High-income countries (>$13,845) are seeking to mitigate the negative impacts of domestic consumption by imposing new sustainability regulations on global supply chains. Recent evidence suggests that these regulations are broadly supported across high-income countries. However, it remains unclear whether citizens of middle-income countries support aligning domestic sustainability regulations with the measures developed by high-income countries. Concerns about economic competitiveness and foreign imposition could increase public resistance toward such alignment. Alternatively, desires for continued market access in high-income countries and aspirations for strengthening local environmental and labour regulations could foster support for alignment. Based on survey-embedded experiments in the three largest democratic non-OECD economies (Brazil, India, Indonesia), we find surprisingly strong support for domestic-based measures that are aligned with emerging global supply chain sustainability regulations. Our findings suggest that support is largely driven by positive impact expectations, where the future benefits of alignment are perceived as outweighing concerns about increased costs. These results bode well for initiatives to install stricter sustainability regulations for global supply chains that are acceptable not only in high-income economies but also in non-OECD countries.

Increased global production practices have driven economic development worldwide[1], but at the same time have also induced substantial environmental (e.g., deforestation, loss of biodiversity) and social (e.g., increased inequality, unsafe working conditions) costs[2,3]. Geographic detachment of production and consumption has allowed higher-income countries (e.g., GNI per capita > $13,845)[4] across the OECD to externalise risks associated with pollution- or labour-intensive production to non-OECD, lower- and middle-income countries (GNI per capita $1136–$13,845)[5,6]. The main enablers of such 'externality-

offshoring' dynamics are trade liberalisation and the associated globalised supply chains (i.e., corporate production networks)[7].

Recently, governance approaches towards mitigating international 'externality-offshoring' have featured more prominently on the political agendas of high-income, OECD countries[8]. In particular, several national governments (e.g., France, Germany) and the European Union are seeking to unilaterally regulate business conduct via 'due diligence'-based legislation, which requires companies to report on sustainability impacts/remedial action throughout their supply

---

[1]ETH Zürich, Zürich, Switzerland. [2]Vienna University of Economics and Business, Vienna, Austria. [3]These authors contributed equally: E. Keith Smith, Dennis Kolcava. ✉e-mail: keith.smith@gess.ethz.ch

chains[9]. These measures aim to further institutionalise and mandate the disclosure-based, best-practice behaviours that have already been widely adopted across a diverse set of sustainability standards initiated by the private sector, such as the Global Reporting Initiative[10,11] and the ISO certification universe[12,13].

However, these first-mover supply chain intervention strategies by governments from high-income, OECD countries generally impose an 'extraterritorial' regulatory framework on production processes outside their authorities' borders. Given the general complexity (e.g., in terms of geographic reach and participating actors) of global supply chains, and given limited resources of 'high-income' government agencies to audit production processes (i.e., actors could refuse compliance), the effectiveness of such interventions will likely depend on the diffusion of such supply chain policy frameworks to prominent production locations and a subsequent harmonisation of the international regulatory environment[14].

Frequently studied sustainability policy diffusion mechanisms from the high-income, OECD states towards non-OECD states often relate to economic 'trading-up' dynamics, caused by companies upward-harmonising company-internal or sector-specific sustainability standards[15]. However, global regulatory harmonisation across various sectors of the economy likely requires coordinated political action[16]. Indeed, many current unilateral supply chain policy efforts by high-income economies already harmonise pre-existing heterogeneous sector-specific sustainability frameworks[17,18]. Therefore, we investigate a crucial political framework condition that could enable or inhibit the spread of cross-sectoral supply chains governance between high-income, OECD states, and non-OECD states.

In particular, we are interested in citizen-level supply chain policy preferences within democratic countries of the high- and medium-income countries. Public opinion has been consistently demonstrated to have a strong influence on governmental actors' priorities and ensuing policy output across wide-ranging democratic contexts[19,20]. Recent findings indicate high support for stringent supply chain policies in high-income OECD countries[21], largely driven by demands to make overseas production practices match ethical expectations[22,23] and the desire to adjust domestic policy to international norm-setting[24].

However, very little is known about public preferences toward global supply chain regulations within middle-income, non-OECD countries: (i) Do citizens in middle-income, non-OECD countries want their governments to implement domestic sustainability regulations to align with the global supply chain measures being developed by high-income, OECD countries? (ii) How stringent would they want these sustainability measures to be? (iii) How do citizens perceive the benefits and cost implications associated with supply chain policies? And lastly, (iv) how do supply chain policy preferences vary across demographic and attitudinal subgroups?

Given substantial variation in economic development, standard theories suggest that sustainability policy preferences within OECD and non-OECD countries might not be aligned. Moreover, in the context of supply chain policy specifically, increased regulatory compliance costs for smaller-scale producers[25] or a new phase of 'imperialist appropriation' appear to further divide preferences[26]. On the contrary, other arguments say that economic development is insufficient to explain the formation of public preferences within highly diverse economically developing societies[27], which, in turn, would allow the alignment of preferences between societies of different income levels.

To address these contradictory predictions, we derive benchmark expectations for global supply chain policy preferences within middle-income states building upon three separate, but non-exclusive, theoretical arguments (please see SI Notes S1–S4 for further elaboration and motivation). First, we expect that public support for supply chain sustainability regulations will be misaligned between high- and middle-income countries – where public support will be lower in middle-income than in high-income countries[28]. Second, citizens in middle-income states will associate larger cost and benefit perceptions with more stringent policy packages[29] – high-stringency designs will have larger perceived benefits (e.g., in terms of improving local production conditions), while at the same time they will be associated with greater perceived costs (e.g., to firms, their employees, and consumers)[30]. Third, individual-level environmental concern and perceived impacts of environmental problems will be positively associated with support for more stringent policies[27]. Lastly, supply chain policy preferences will be robust against information provided about potential costs and benefits associated with these measures[21].

In this work, we test these expectations utilising survey-embedded experimental designs to explore preferences amongst the three largest, democratic non-OECD economies (Brazil, Indonesia and India). We find, relatively surprisingly, support for the development of domestic-based policy measures in alignment with emerging global supply chain sustainability regulations. Within Brazil, Indonesia and India, a majority of respondents support medium- and high-stringency policy designs. Positive citizen evaluations are broadly shaped by greater perceived policy benefits than by concerns over cost implications. We further find substantial evidence of cross-national heterogeneities, where support is comparatively higher within Brazil and India than in Indonesia. These findings suggest that an alignment towards stricter sustainability regulations for global supply chains is acceptable not only in high-income economies but also in non-OECD countries.

## Results
### Research design
We study public opinion towards global supply chain policies in three large, middle-income democracies: Brazil, India, and Indonesia. Specifically, we utilise multiple original survey-embedded experiments to identify the acceptability of sustainable global supply chain regulations across the three largest non-OECD democratic economies, and in comparison with the 12 largest OECD importing, democratic economies (BE, CA, CH, DE, ES, FR, IT, JA, KO, NL, UK, US). First, based on a conjoint choice experiment, we assess the acceptability of policy designs across regulatory stringency levels (low, medium, high) based on variation in policy scope, transparency requirements, and enforcement capacity in non-OECD and OECD states. Next, we use a vignette-experimental design to identify perceptions of policy benefits and costs across regulatory stringency within non-OECD contexts. Third, we explore the acceptability of policy designs across subgroups, and lastly, assess the robustness of citizens' policy preferences using a multidimensional informational treatment design.

To identify the acceptability of global supply chain policy instruments, we first use an original conjoint choice experiment[31], presenting the results for Brazil, Indonesia, and India, the world's largest (GDP) democratic non-OECD economies (see Table 1). These three countries are classified by the World Bank Country and Lending Groups as middle-income economies: with Brazil and Indonesia being classified as 'upper-middle-income' ($4466–$13,845 GNI per capita) and India as 'lower-middle-income' economy ($1136–$4465 GNI per capita)[4].

By exploring variation within and between these large middle-income countries, our findings extend recent work on the acceptability of supply chain policy instruments across the OECD[21]. We discuss the similarities and differences between these studies in detail within the Methods section.

Here, we develop four distinct analyses, each matching an expectation developed above. First, we utilised a conjoint experimental design[31] to assess support for supply chain preferences by policy design elements in Brazil, India, and Indonesia. The design is implemented as a parallel study to an experiment fielded across 12 OECD countries[21], where the primary difference is the framing of

**Table 1 | Overview of middle-income country characteristics**

| Dimension | Brazil | India | Indonesia |
|---|---|---|---|
| GDP (IMF, nominal, bn. US$) | $1894 bn. | $3468 bn. | $1289 bn. |
| GDP per capita (IMF, PPP, US$) | $17,684 | $8293 | $14,638 |
| Economist Democracy Index (1–10, democratic > 6) | 6.86 | 6.91 | 6.71 |
| FDI as % of GDP (OECD, 2022[49]) | 1.3% | 0.6% | 0.3% |
| FDI Restrictiveness Index (OECD, 2020[38]) | 0.081 | 0.207 | 0.347 |
| Top 3 Exports (OEC) | Soybeans, iron ore, crude petroleum | Refined petroleum, packaged medicaments, diamonds | Palm oil, coal briquettes, gold |
| Main Export Destinations (OEC, bn. US$) | China $88.3 bn., United States $30.2 bn. | United States $71.2 bn., United Arab Emirates $25.4 bn. | China $54.5, United States $26.2 bn. |

**Table 2 | Policy stringency scenarios**

| Stringency | Scope | Transparency | Enforcement |
|---|---|---|---|
| Low | The law applies to very large companies with 25,000 or more employees. | Companies write an annual confidential report to the government. There are no government rules on required content (companies can freely choose what they report). | If a company withholds or presents false information the government cannot take any action against the company. |
| Medium | The law applies to large and medium-sized companies with 250 or more employees. | Companies write an annual confidential report to the government. There are some general government rules on required content (companies can only partially choose what they report). | If a company withholds or presents false information the government can put the company on a public list of companies that provide unreliable information and impose a moderate financial penalty. |
| High | The law apply to all companies with 25 or more employees. | Companies write an annual public (online) report. There are detailed government rules on required content (companies must report all required information). | If a company withholds or presents false information the government can put the company on a public list of companies that provide unreliable information, impose a severe financial penalty, stop buying government supplies from that company and press legal charges against the company management. |

preferences towards aligning domestic policies with measures being developed across the OECD (see Table S1 in Supplementary Information for comparison). The conjoint attribute and levels are identical for both experiments. The conjoint contains three attributes (scope, transparency and enforcement, see Table S2 in Supplementary Information for comparison), where the combination of all levels produces a full factorial of 36 distinct policy designs. For parsimony, we adopt three stylised designs for our analyses, representing high, medium, and low-stringency packages (see Table 2).

The conjoint experiment is evaluated using marginal means[32] and summarised the results in Fig. 1. Marginal means can be interpreted as the predicted probability of supporting a global supply chain policy proposal in a referendum or popular vote. We first identify average levels of support across all policy instruments for each of these regions in Fig. 1A. Next, we evaluate support by the stylised low-, medium-, and high-stringency policy packages in Fig. 1B. The full marginal means of public support for all policy instruments reported in Figs. S1 and S2.

Second, we utilise a vignette-experimental treatment design to identify the perceptions of benefit and cost implications resulting from the sustainable supply chain proposal, across low, medium, and high-stringency levels. Here, we randomly assign respondents to evaluate the perceived implications of one of three randomly assigned low-, medium-, and high-regulatory stringency treatment packages. Benefits are assessed as facilitating better consumer information availability, production conditions, and creation of domestic jobs. While we assess the expectations of costs these policies would impose on firms, consumers and affect domestic sovereignty (see Table S3 in Supplementary Information).

We calculate marginal means for each of the benefit and cost perceptions by regulatory stringency package in Brazil, Indonesia, and India in Fig. 2. Greater marginal means can be interpreted as higher expected benefits/costs implications, given a particular regulatory stringency package (Likert-scale outcome, ranging from 1 'strongly disagree' to 7 'strongly agree').

Third, we further evaluate the conjoint experimental design, conditioning these by four respondent characteristics in Fig. 3. We explore how support for supply chain policies across regulatory stringency levels (low, medium, high) is conditioned by environmental attitudes, perceived environmental impact, education, and household income. As in Fig. 1, we calculate the marginal means of support for supply chain policies by policy stringency in Brazil, India, Indonesia conditional on the subgroup characteristics.

Lastly, we evaluate how sustainable supply chain preferences can be conditioned by exposure to potential future costs and benefits resulting from these policy proposals. Directly before evaluating the proposals in the conjoint experiment, respondents are randomly presented with 1–5 treatment allocations (see Table S4 in Supplemental Information). The potential treatment allocations are two statements regarding the potential benefits of these policy proposals (improved conditions and increased trade and jobs), two statements on potential costs (to consumers and jobs loss, and threats to sovereignty), and one control condition (where the respondent is not presented with a statement). Here, we calculate marginal means of support for supply chain policies by policy stringency in Brazil, India, and Indonesia conditional on the treatment allocation in Fig. 4. Potential manipulation by experimental information allocation can be observed by statistically significant differences in policy support (e.g., nonoverlapping 95% confidence intervals) within each level of policy stringency.

## High acceptability of global supply chain policy packages

Using data from the conjoint choice experiment, Fig. 1A displays the support for sustainability regulation policies averaged across all policy instrument conditions. We find similarly strong levels of support for supply chain policies in Brazil (64.5%), India (64.1%), and in the 12 OECD countries. In contrast, within Indonesia, only a narrow majority (50.9%) of supply chain policy proposals are supported.

Next, turning to how support is conditioned by levels of policy stringency in Fig. 1B, we find substantial evidence of support for

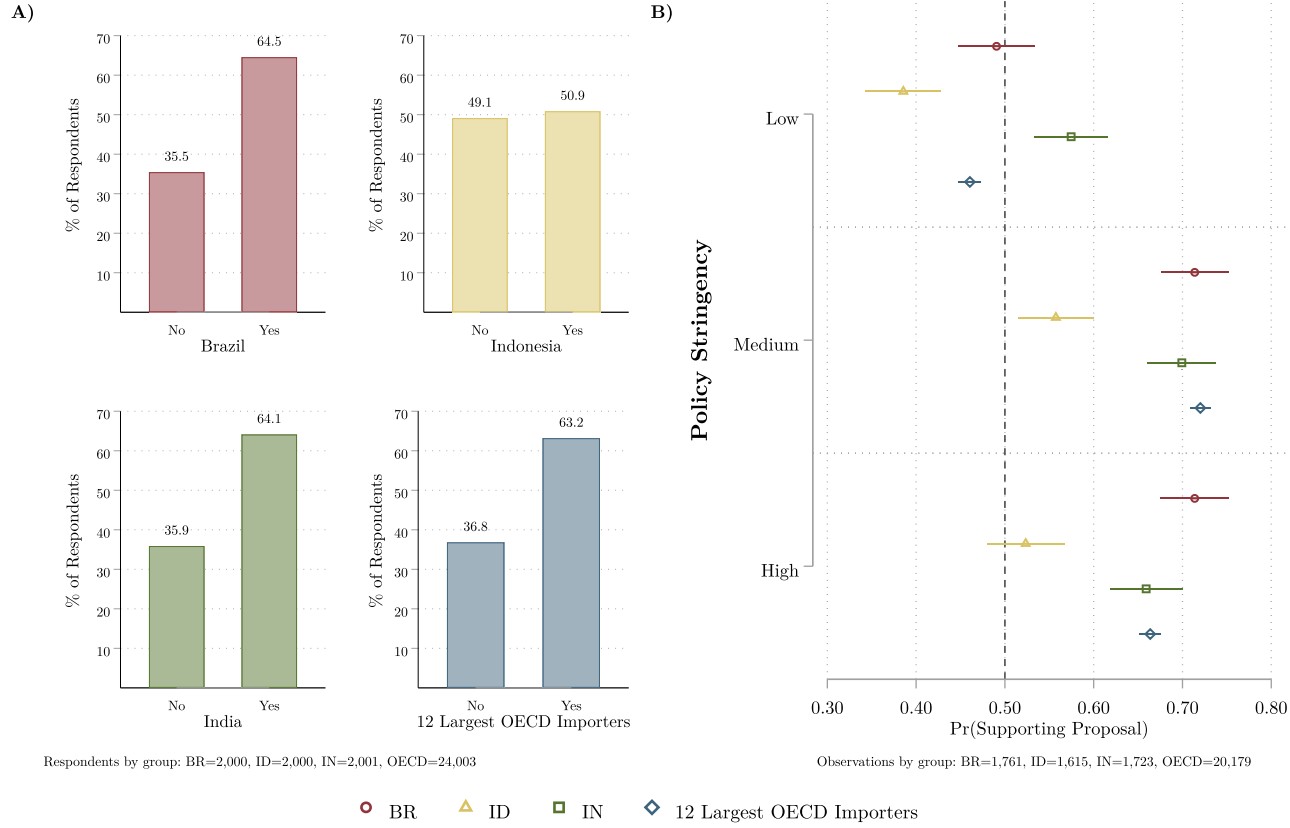

Respondents by group: BR=2,000, ID=2,000, IN=2,001, OECD=24,003

Observations by group: BR=1,761, ID=1,615, IN=1,723, OECD=20,179

○ BR △ ID □ IN ◇ 12 Largest OECD Importers

**Fig. 1 | Support for global supply chain policies in BR, ID, IN, and the 12 largest OECD countries (by imports).** The estimates for Brazil are plotted in red, for Indonesia in yellow, for India in green, and in blue for the 12 largest OECD economies. **A** The average level of support across all policy treatment conditions for each region. **B** The marginal means of predicted support for policies by regulatory stringency for each region, with 95% confidence intervals. Both panels use data from choice-experimental survey designs.

policies with 'medium' and 'high' levels of regulatory conditions within Brazil, India, and the 12 largest OECD importers. Notably, across these three regions, approximately 65–70% of the public supports high-stringency global supply chain policies – those which would apply to all companies with 25 or more employees (scope), require annual, detailed, mandatory public information disclosure (transparency), and in cases of non-compliance, allow for governments to publicly name offending firms, impose severe financial penalties, and press legal charges against management (enforcement) (see Table 2 for the detailed in-survey descriptions of policy stringency). Among Indonesian respondents, policy packages with 'medium' and 'high' levels of regulatory stringency garner comparatively lower, but still a majority of support (50–55%).

Overall, the 'low' stringency package has the least public acceptability. Low regulatory stringency does not garner a majority of support in three of the study regions – only 39% of the respondents in Indonesia replied that they would vote for this policy design, compared to 46% in the OECD countries and 49% in Brazil.

In sum, these findings suggest that, in response to the increase in global supply chain sustainability regulations in the OECD, there is broad acceptability for stringent, sustainability-orientated global supply chain policies within Brazil and India. Furthermore, while the supply chain policies are comparatively less supported in Indonesia, a plurality of citizens are accepting of medium- and high-stringency policy packages.

### Policy benefit expectations are greater than perceived cost consequences

Drawing upon the vignette-experimental design, we assess the perceptions of benefits (better information provision to consumers, safer

and more environment-friendly local production practices, and create domestic jobs) and cost implications (increase economic costs for firms, consumers, and impose state-level opportunity costs by restricting national sovereignty) associated with varying levels of supply chain policy regulation stringency (Fig. 2).

Across all three middle-income countries, people are generally more likely to have greater expectations for the benefits associated with supply chain policies rather than expectations for costs, regardless of the level of regulatory stringency. First, the overall marginal means for all three indicators of benefit perceptions is substantively high (between 5 'somewhat agree' to 6 'agree'). In particular, for Indonesia and India, there is little differentiation in benefit perceptions across regulatory stringency levels. Meanwhile, for Brazilian respondents, perceptions of informational, production, and job creation benefits are slightly less for the low-stringency design (marginal means ranging from 5.0 to 5.3) than in comparison with perceptions of the high-stringency design (5.7–5.9).

Moreover, we observe differential cost perceptions by country. Respondents in India are most likely to have higher cost expectations associated with supply chain policies, regardless of regulatory stringency (marginal means ranging from 5.0 to 5.5). In comparison, cost expectations for firms and consumers are comparatively lower in Brazil across stringency packages (marginal means ranging from 3.6 to 3.8). Notably, we find little evidence of a conditional effect of regulatory stringency on cost expectations associated with particular policy packages (except for the case of national sovereignty in Brazil). Rather, cost perceptions appear to vary more by country context than by experimentally manipulated levels of regulatory stringency.

Summarily, we find that expected policy benefits outweigh expected policy costs across all three middle-income countries.

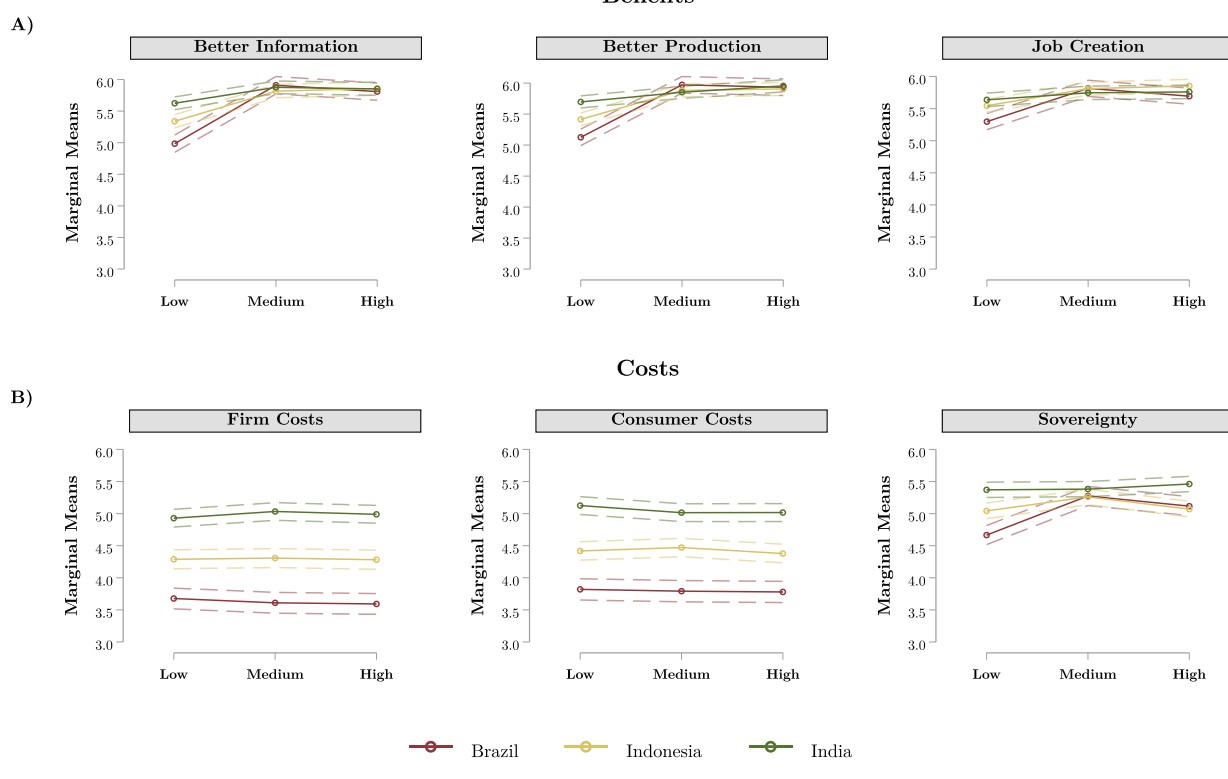

Observations by country: BR=2,000, ID=2,000, IN=2,001

**Fig. 2 | Perceptions of benefits and cost associated with global supply chain policies by regulatory stringency for BR, ID, and IN. A** Perceptions regarding the benefits of global supply chain policies (better information, better production, and job creation) for the three middle-income countries. **B** Perceptions of costs associated with global supply chain policies (firm costs, consumer costs, and sovereignty). Marginal means of perceptions are plotted by experimentally assigned regulatory stringency packages, with 95% confidence intervals plotted in dashed lines. The marginal means for Brazil are plotted in red, for Indonesia in yellow, and for India in green.

Indeed, within any of these three countries, we find no cases where the marginal means of the cost perception indicators exceed those for benefits given a specific regulatory stringency level.

### Policy support varies by subgroup characteristics

In order to further evaluate supply chain policy preference formation, we stratify the conjoint experimental findings by individual characteristics for the three middle-income countries. Figure 3 reports our findings suggesting substantial differentiation in the drivers of policy support.

First, within Brazil (Panel A), supply chain policy preferences are not strongly conditioned by any of these four individual characteristics and attitudes. Support for high- and low- stringency policies increases moderately with heightened environmental attitudes and perceived exposure to environmental impacts. While support for high and medium-stringency packages decreases slightly with increased education and household income. But overall, the substantive sizes of these conditional effects are rather small (-5% differences in support).

Next, turning to Indonesia (Panel B), we find that policy preferences are strongly moderated by educational attainment. Indonesians with high levels of education (e.g., 15–20 years) are very likely (60–70%) to support high-, and medium-stringency policy proposals. Furthermore, Indonesian respondents with heightened environmental attitudes and perceived exposure to environmental impacts are substantively more likely to support high-stringency policy packages (20–25% more likely). Yet, we comparatively do not find as much variance in support by household income.

Third, within India (Panel C) we find similar patterns of conditioning by subgroup characteristics as those observed in Indonesia. Here, support for medium- and high-stringency policy packages greatly increases at higher levels of educational attainment. Furthermore, we find that support for all policy proposals, at low-, medium- and high-stringency levels, increases at higher levels of environmental attitudes and perceived exposure to environmental impacts. Lastly, we again find smaller conditioning of support by household income.

Taken together, these findings suggest that support for supply chain policy support is driven by heterogeneous factors across large middle-income democracies. We find substantial differences across subgroups within Indonesia and India. Heightened education, environmental attitudes, and perceptions of environmental impacts are associated with increased support for high-stringency policy packages in Indonesia and India. Yet, for Brazil, we find little evidence of substantial conditioning of policy support within these subgroups. Although in all countries we find less evidence of conditioning by household income levels.

### Responsiveness to benefit and cost information

Lastly, we identify the resiliency of supply chain policy preferences against five randomly assigned informational treatment allocations. Fig. 4 summarises our pooled analysis of respondent responsiveness to information on the benefits and costs of global supply chain policies in Brazil, Indonesia, and India by policy stringency packages. Relative to policy preferences in the control group (no information is provided), we did not find statistically significant differences in the levels of support for each of the three policy stringency packages by any of the

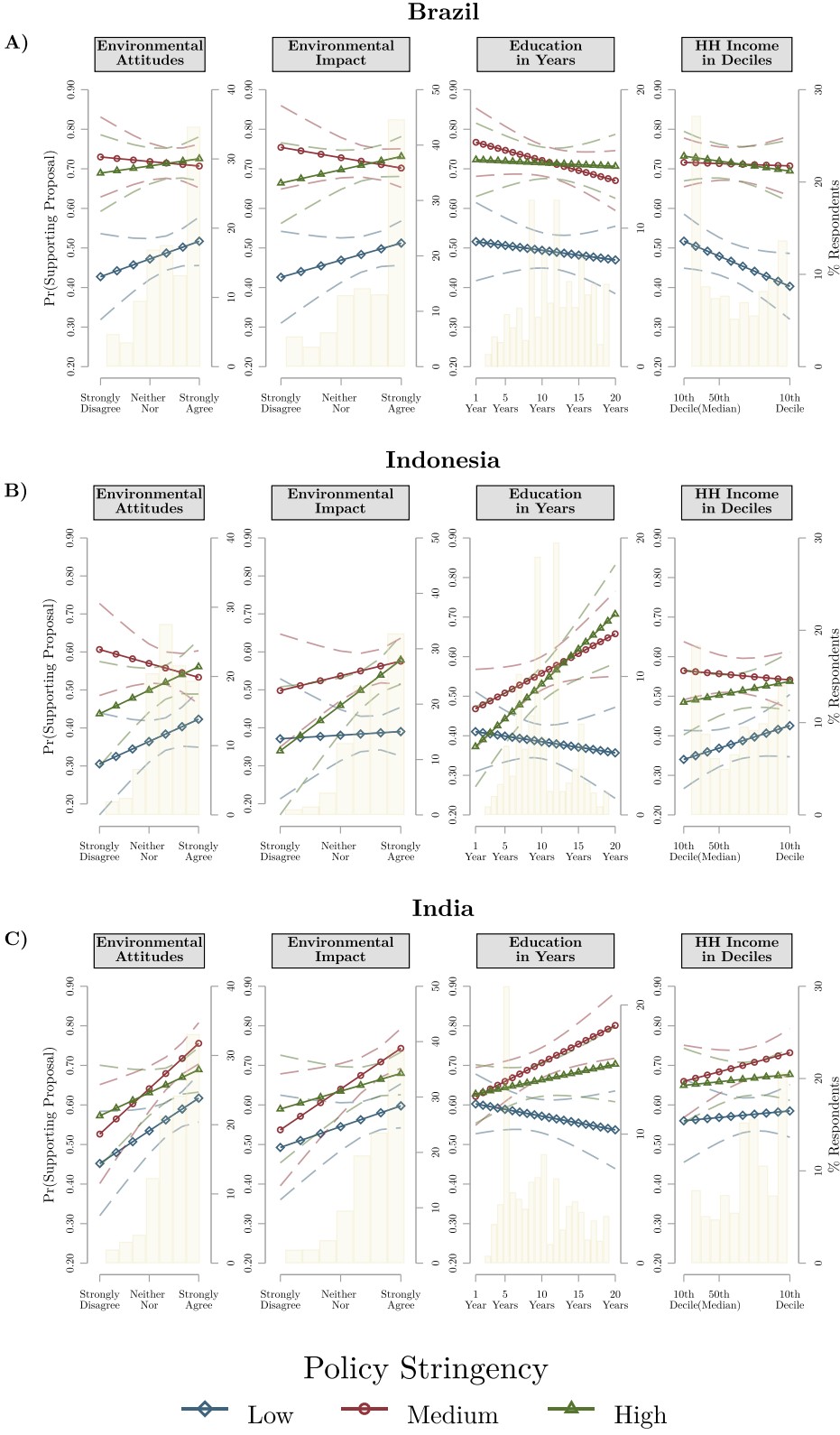

**Fig. 3 | Support for global supply chain policies in BR, ID, IN by subgroup characteristics. A** Support for global supply chain policies by low, medium, and high policy stringency packages across the respondents' levels of environmental attitudes, perceived environmental impact, education in years, and household income in deciles for Brazil. **B** The same information for respondents in Indonesia. **C** The same information for respondents in India. The marginal means of predicted support (left y-axis) for policies by regulatory stringency are plotted for each region, with 95% confidence intervals in dashes by different subgroup characteristics. High-regulatory stringency is plotted in green (triangle), medium stringency in red (circle) and low stringency in blue (circle). Marginal means are calculated at substantive levels of the subgroup characteristics. The distributions of subgroup characteristics by percentage of respondents (right y-axis) are plotted in yellow for each country.

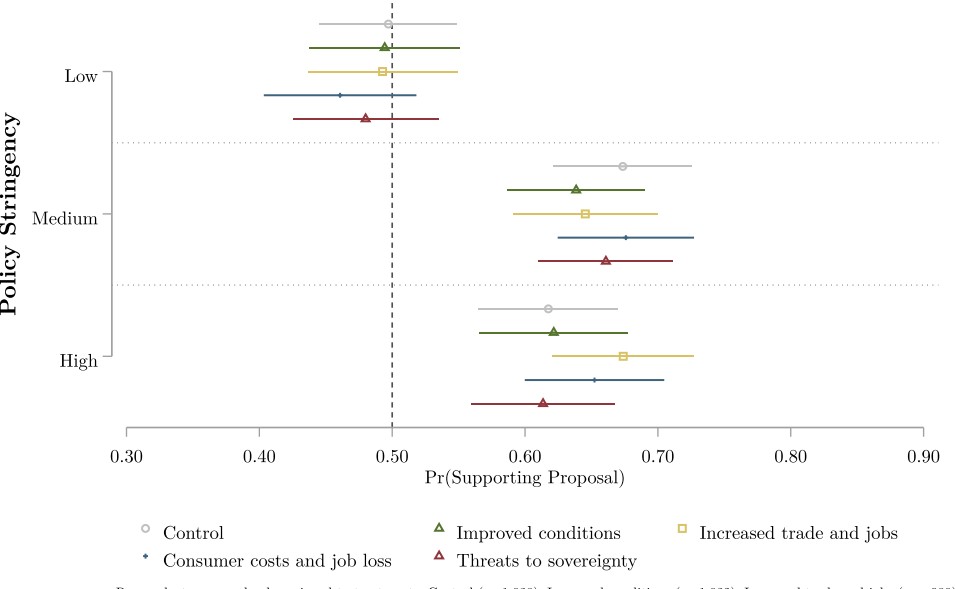

Respondents are randomly assigned to treatments: Control (n=1,038), Improved conditions (n=1,032), Increased trade and jobs (n= 980), Consumer costs and job loss (n=1,015), and Threats to sovereignty (n=1,034) for a total of (n=5,099) observations

**Fig. 4 | Pooled support for global supply chain policies by exposure to informational treatment statements.** Data is pooled across samples from Brazil, Indonesia, and India. The marginal means of predicted support for policies by regulatory stringency are plotted with 95% confidence intervals by exposure to information treatment. Informational treatments are randomly assigned to respondents: control (no information) is plotted in grey (circle), improved conditions are plotted in green (triangle), increased trade and jobs are plotted in yellow (square), consumer costs and job loss are plotted in blue (plus), and threats to sovereignty is plotted in red (diamond).

benefit or cost informational statements. We interpret these results as evidence that preferences towards sustainable supply chain policies in Brazil, India, and Indonesia are largely robust against informational provisions.

## Discussion

Stringent sustainability regulations on global supply chains could help facilitate stark reductions to inequalities in pollution- and labour-risk-distributions between higher- and middle-income countries[33]. Based on survey-experimental methods, we explore how support for sustainability regulations on global supply chains varies with regard to (i) cross-national alignment and policy stringency, (ii) benefit and cost perceptions, and (iii) individual characteristics.

First, we find that citizens of non-OECD, middle-income countries are supportive of domestic policies which align with emerging extra-territorial supply chain measures developed across OECD states, particularly at higher levels of policy stringency. The overall levels of support vary cross-nationally: Our findings in Brazil and India suggest substantial public support for aligning domestic policies with current regulatory requirements developed across OECD countries. While policy preferences towards alignment are comparatively lower in Indonesia, a majority of respondents support medium- and high-stringency policy designs. Accordingly, these findings suggest that if higher-income, OECD countries place greater sustainability requirements on imported products, citizens within Brazil and India are likely to be supportive of domestic government regulation of production conditions to ensure compliance.

Furthermore, these findings alleviate frequently voiced concerns (e.g., in trade policy negotiations) that an alignment of domestic sustainability regulations with regulatory mandates from higher-income states could be preceived as a threat to economic development, and rather, corroborate research demonstrating strong support for establishing governance structures at the intersection of globalisation (e.g., trade) and sustainability in non-OECD countries[34,35]. Furthermore, these findings are somewhat at odds with classical macroeconomic expectations of preferences for reduced regulatory capacity within middle-income states, and rather support suggestions that citizen may be more strongly in favour of environmental and labour protections. Accordingly, citizen demands could contrast with those of domestic policymakers[36], which are often presumed to oppose foreign-based impositions and environmental protections.

Second, we find that, on average, citizens across the three non-OECD, middle-income countries perceive that the sustainable supply chain policies will have comparatively greater benefits (better information, production practices, job creation) than cost implications (to firms, consumers, and to national sovereignty). These findings are consistent across Brazil, India, and Indonesia, as well as across levels of policy stringency. Hence, while ex-post impact analyses of voluntary sustainability governance initiatives suggest limited effectiveness[37], the public in these countries rather expects that transparency regulations could improve information, production practices, and create local jobs. As perceived benefits do not substantially vary across regulatory stringency packages, these expectations may reflect a desire by citizens to just 'do something' in this rather weakly regulated policy domain.

Third, we find substantial conditioning of support by subgroup characteristics in India and Indonesia. Respondents with greater levels of environmental attitudes and impact perceptions and educational attainment are more likely to support stringent sustainable supply chain policy designs. These findings are in alignment with expectations about the role of environmental attitudes in shaping policy preferences. Yet, notably, we find minimal differences across these subgroups in Brazil. We interpret these findings as evidence of further cross-national heterogeneity within non-OECD countries, discussed further below.

While we find higher levels of overall support for sustainable supply chain policies in Brazil (65% support across all conditions), fitting with expectations, policy support is comparatively higher in less affluent India (64%) than in Indonesia (51%). These findings are somewhat in contrast to our prior expectations. A macroeconomic approach would suggest that new measures regulating economic activity could be seen as a restriction on development, particularly in

non-OECD, producing states. Yet, we find a majority of citizens support aligning domestic policies with OECD supply chain regulations, particularly within India. These findings suggest the potential role of alternative macro-structural conditions beyond GDP per capita, such as differential opportunity costs due to countries' (sustainability) policy misalignment with their main export markets (the US for India, China for Indonesia). In a similar vein, citizen preferences toward aligning sustainable supply chain policies along OECD benchmarks could be shaped by the magnitude of FDI in the domestic economy. In particular, among the countries included in the OECD FDI Regulatory Restrictiveness Index[38], Indonesia has one of the greatest sets of restrictions for foreign investment and development worldwide – only exceeded by the Philippines, Palestine, and Libya. As a result, FDI plays a comparatively limited economic role in Indonesia, and thus the opportunity costs of sustainability policy misalignment with potential OECD investor economies may be smaller.

Furthermore, we explored conditions shaping preference formation, noting that citizens perceive greater potential benefits than cost implications resulting from these proposed sustainable supply chain measures. Such findings are in congruence with heightened levels of overall support. We also found that subgroup conditions shape support in India and Indonesia – where increased preferences for sustainable supply chain policies are driven by pro-environmental attitudes and concerns. Yes, these findings do not fully explain the variation in regulatory preferences between India and Indonesia.

We conducted further empirical analyses of the conjoint experiment to explore data-driven explanations for these differences. Broadening beyond the stylised low, medium, and high policy packages, we explore support for the 'full factorial' of policy package conditions (see Fig. S2 in Supplementary Information). Here we find that respondents in Indonesia are similarly likely to support high-stringency policy designs as people in Brazil and India if these are applied to larger firms only (e.g., ≥25,000 employees). While, once the regulations are applied to smaller firms (e.g., ≥25 employees), support drops by 10–15%. Accordingly, citizens in Indonesia may still support more stringent policy packages, yet have concerns about the effects on smaller-scale producers.

In sum, these preliminary, cross-national findings strongly suggest further country-specific studies to explore patterns of within-country variation in supply chain policy preference formation. Notably, exploring individual factors (such as employment sector, respondent region, and attitudinal dispositions) and further systematic inquiry into macro-level drivers.

Taken together, the 15 democratic countries covered in our study account for ~35% of the global population and ~55% of the global GDP. Therefore, the overarching alignment of public preferences on stringent sustainability regulations for supply chains across middle- and high-income (OECD) countries has important consequences for global sustainability governance. In particular, the alignment of public attitudes could be an enabling condition for the diffusion of policy regimes along economic links from the high-income, OECD countries to middle-income, non-OECD states – in some ways, an individual-level variation of more economic-structural 'trading-up' arguments[39]. As a result, there appears to be a window of opportunity for cross-national policy sequencing processes upon the momentum of current policy action among OECD economies and the support these policies garnered in corresponding debates[40].

Against this backdrop, we derive three specific recommendations for policy action: First, international policy and economic actors with leverage should focus on pressuring national governments in countries such as Brazil, India, or Indonesia to implement disclosure-based policies for extraction and production processes that feed into global supply chains, thereby contributing to a harmonisation of the international regulatory environment.

Furthermore, second, policymakers should focus not only on increasing the stringency of transparency regulations (such as the required level of detail for corporate reports) but rather maximising the value of the disclosed information within diverse stakeholders' decision-making. This could, for instance, mean pushing the geographic coverage of disclosure adoption within geographically large and highly diverse middle-income countries (such as Brazil, India, and Indonesia) by implementing standardised reporting criteria to allow for low-cost compliance and ensure the comparability of production sites' performance[41].

Lastly, a successful harmonisation of international supply chain governance likely entails ancillary measures to balance out inequities in stakeholders' capabilities to comply with new rules[25]. This is essential as non-compliance due to capability deficits is likely to occur among economically vulnerable actors (for example, small-scale farmers and miners), whose environmental and work conditions could deteriorate if their access to global markets is reduced[41] – the exact opposite of what supply chains transparency policies aim to achieve.

In sum, our findings bode well both from a normative viewpoint (i.e., the alignment of public preferences in OECD, and non-OECD countries) and with regard to the feasibility of policy implementation. However, since this study focuses on three large middle-income democracies, future research in middle-income settings might provide a stronger probe into preference formation toward sustainability regulation. This is particularly important given that, while our findings for OECD and non-OECD contexts provide robust evidence of cross-national public policy preference alignment, the role of particular opinion drivers (e.g., economic gains, normative elite cues) within individual preference formation could be explored further. Moreover, while we explore how perceptions vary across income levels within each country, the distributional impact of these policies further likely varies substantially within each of these settings. Given the cross-national design, we are unable to identify perceptions of subgroups most likely to be affected by these policies. This is a limitation of these exploratory findings, and we suggest further research into the distributional impacts of sustainable supply chain policy proposals within these locales as an extension of these initial findings. Lastly, while we adopted a quota-based sampling technique, we faced the common difficulty of recruiting respondents with lower education levels in Brazil. We suggest further research adopting probability-based sample designs to further evaluate the generalisability of these initial findings.

In a similar vein, given that the translation of public preferences into regulatory policy may be conditional on structural factors (e.g., the presence of parochial interest group pressure), further research in middle-income countries could assess potential disconnects between public and elite preferences on the governance of sustainability in production processes. Lastly, while public opinion is presumably less effective in driving public policy choices in autocratic states, autocratic policymakers may still be responsive to citizens' interests[20] and demands for regulatory enforcement in environmental and labour policy[42] – which could be investigated in the context of multinationals' subsidiaries.

## Methods
### Data collection
We implemented an original survey instrument (April 27 and May 23, 2022) in Brazil, India, and Indonesia (N = 2000 each, 6000 total), the three largest (in terms of nominal GDP) democratic non-OECD economies (see Table 1). We sampled from voting-age (18) citizens within Dynata's online panel, adopting representative quotas on age, gender (interlocked with age), and education (3 categories), see Table S5 in Supplementary Information. The recruitment, data collection and storage, and survey instrument were approved by the ETH-Zurich Ethics Commission (EK-2021-N67). The study design was pre-registered on the Open Science Framework (see: https://doi.org/10.17605/OSF.IO/7ATUP).

The survey instrument was fielded in the main official languages of Brazil, India, and Indonesia: English, Hindi, Indonesian and Portuguese. We relied on a two-step translation procedure. First, the survey instrument was professionally translated from the English original version into all other languages. Second, we revised each of the translations with input from native-speaking social scientists to insure proper translation of technical concepts. We further implemented three attention checks within the survey instrument: (i) response duration – threshold of 45% of the median duration (7 min, 3 s), (ii) item indicating how many wheels a bicycle has, and (iii) cross-checking whether the respondent's stated age (beginning of survey) matches their year of birth (end of survey). Respondents that failed at least 2 of these 3 checks were excluded from the final analytical sample (roughly 3% of the sample). Table S6 in Supplementary Information presents full attention check item information. Lastly, to incentivise participant engagement with the survey instruments, we disabled the "next" button on key information pages/treatments for several seconds depending on the text length displayed.

## Supply chain policy preferences

Within our instrument, we implement three separate original survey-experimental designs. First, we adopt a conjoint choice experiment[43] to measure citizen supply chain policy preferences. In the experiment, participants are tasked with assessing two policy packages, Proposal A and Proposal B, which are displayed next to each other. The packages consist of three policy design attributes (scope, transparency requirements and enforcement actions), operationalising distinct dimensions of supply chain policy. Further, each attribute contains varying levels of policy instruments. The policy packages displayed to participants contain one randomly drawn level (instrument) for each attribute (policy dimension), summarised in Supplementary Information Table S2. Given the full-factorial research design, respondents are randomly displayed packages generated from 36 distinct policy designs. Each respondent participated in five separate rounds of the conjoint experiment, therefore, comparatively evaluating 10 policy packages.

Further, we draw upon data from a previous original survey instrument (September 22 and November 3, 2021) implemented in the 12 largest importers within the OECD – BE, CA, CH, DE, ES, FR, IT, JA, KO, NL, UK, US, n-2000 each for a total of $n = 24{,}003$. Here we also recruited respondents from Dynata's online panel, utilising an identical strategy for age (18+) and quotas (age, gender (interlocked with age), and education (3 categories)). The survey instrument was fielded in English (for all countries), as well as the main official languages for each target country, including German, French, Italian, Spanish, Flemish, Dutch, Japanese, and Korean. Here, we similarly utilised a process of professional translations, which were reviewed by native speakers to assure accuracy across languages. The survey instrument was approved by the ETH-Zurich's Ethics Committee (EK-2021-N67), and was pre-registered at the Harvard Dataverse (see: https://doi.org/10.7910/DVN/KLOTB6).

We also implemented attention checks within this survey data collection, which flagged respondents across the following criteria: (a) response duration below 45% of the median duration (b) incorrect response to how many wheels a bicycle has (c) and incorrect answer on an item requesting respondents to 'select the triangle'. Respondents which failed at least two of these three items were replaced in the sampling process, and therefore excluded from empirical analysis (~3.2%).

Unless noted and described in greater detail in the "Methods" section, all aspects of the survey design and instrument were harmonised between middle-income and high-income OECD states to ensure comparability.

Supply chain policy proposals are modelled across three dimensions of policy attributes: scope, transparency, and enforcement. First, 'scope' is an indicator of regulatory breadth[44], outlining which firms would be required to comply with the potential new regulations (denoted by varying number of employees). Here, we consider the lower employee thresholds to represent higher levels of policy stringency. Second, 'transparency' describes the sustainability information disclosure requirements to which firms are subjected. Thereby, requirements vary on two sub-dimensions: the degree to which the companies disclosure report is public or not, and government-mandated versus firm-led reporting components. With regard to policy stringency, we consider policy packages with public reporting requirements as well as mandatory reporting components to be more stringent. Lastly, 'enforcement' represents the capacity for governmental sanctioning in cases of non-compliance with required disclosure activities. The lowest stringency level comprises no enforcement capacity, while subsequent levels increased stringency along multiple sub-dimensions – 'naming and shaming', imposing fines and legal action against firm executives.

For parsimony, we construct three policy packages representative of 'low', 'medium', and 'high' regulatory stringency designs (see Table 2). For example, the low regulatory stringency package consists of the lowest levels for scope (25,000 employees or more), transparency (confidential report with no required content), and enforcement (government cannot punish non-compliance). On the contrary, the high-regulatory stringency package contains the highest levels for scope (firms with 25 employees or more), transparency (annual public report with detailed required content), and enforcement (naming and shaming, high fines, potential legal action). Accordingly, these policy packages represent three ideal types of regulatory stringency levels, drawn from the full factorial of 36 potential policy designs.

Conjoint experimental designs commonly adopt a 'forced-choice' response as the key measure for policy design acceptability, where respondents are tasked with evaluating the two proposals on display in comparison, and selecting which of the two they would prefer (Proposal A or Proposal B). We expand upon this methodology to incorporate two further evaluation measures.

First, before the 'forced-choice' item, respondents are asked whether they would support Proposal A and B (separately, yes/no) in a hypothetical national vote (e.g., referendum or initiative). This 'proposal support' approach has several advantages. Under forced-choice evaluations, the researcher is able to compare acceptability between different policy instruments (levels). Yet, the absolute level of policy support is obfuscated in forced-choice designs, as responses are centred around an arbitrary 50% mean. This presents a notable limitation, as the absolute level of policy support carries at least equal substantive value as the relative measure of policy support. While the 'proposal support' measure allows for comparative evaluation across policy instruments (levels) to be made alongside substantive, absolute levels of support (e.g., 55% of proposals containing an instrument are supported). The 'proposal support' methodology has been adopted in recent policy acceptability analyses[45], and is preferable over common rating evaluations, which often adopt Likert-scale outcomes that are difficult to interpret substantively[46].

Second, in a 'proposal support' situation, if participants indicated that they are in opposition to both proposals, we prompted them with a follow-up, in which we asked participants if their opposition is because 'neither package was strict enough', 'both packages were too restrictive', or another reason, which they could enter in an open text field. Accordingly, this 'proposal opposition' measure complements measures of policy design support, allowing for a more complete understanding of why participants find a design unacceptable.

## Preference formation

In order to evaluate patterns of preference formation towards sustainability in supply chain policy designs, we adopt a second survey-experimental design using a vignette methodology. Directly after completing the conjoint experimental components, participants are

asked to evaluate a policy design with regard to its perceived benefits and costs. Participants are randomly assigned one of the 'low', 'medium', and 'high' regulatory stringency designs (see Table 2), and then, asked to evaluate the package across statements of resulting benefits (improved information availability, improved production conditions, stronger domestic job creation), as well as potentially resulting costs (firm costs, consumer costs, loss of national sovereignty).

Participants are asked to evaluate each of these benefit and cost perception statements (see Table S3 in Supplementary Information) on a Likert-scale, ranging from 1 'strongly disagree' to 7 'strongly agree'. Furthermore, we randomly vary whether improvements in consumer sustainability information and production processes will accrue in terms of improved 'working' or 'environmental' conditions. In supplementary analyses, we did not find statistically significant differences in benefit perceptions across the two sustainability domains (see Fig. S3 in Supplementary Information), and accordingly, we present the pooled results across these randomly assigned attributes.

### Acceptability by subgroup characteristics

First, we evaluate acceptability between citizens of non-OECD and OECD democratic economies. Here we draw upon additional survey data of 24,003 participants from the 12 largest high-income importing economies (BE, CA, CH, DE, ES, FR, IT, JA, KO, NL, UK, US, $N = 2000$ each)[21]. The conjoint experimental instrument is identical for both the non-OECD and OECD contexts. In particular, both designs are based on policy packages drawn from identical policy attributes, which varied at random between identical values in terms of wording. Furthermore, the proposal evaluations item wordings (i.e., 'forced-choice' and 'policy support') are also identical.

There are minor differences in the conjoint experiment design introduction, due to the two divergent contextual perspectives on global supply chains – that is, a 'consumer-based, industrialised' perspective and a 'producer-based, lower- and middle-income country' perspective. Accordingly, the introductory information in the non-OECD refers to implementing policy measures towards *local production of export goods*, introductory information within the OECD survey was framed as *local consumption of imported goods*. We report the introductory information in Table S1 in Supplementary Information.

Specifically, step 1 introduces the international economic linkages between countries in both surveys but reflects the different (consumption vs production) perspectives on global supply chains. As a result, in the non-OECD survey, the policies are domestic (i.e., regulating production processes within country X), whereas in the OECD survey they are extraterritorial (i.e., regulating production processes of products that are eventually sold within country X regardless of the production location). Nonetheless, the policy instruments that are evaluated remain the same. Thus, ultimately, the resulting differences between the OECD and the non-OECD surveys pertain to the implementation location of the regulation, not the elements of the regulation itself. Step 2 of the introductory information is virtually identical in terms of providing information about differences in production regulations between countries. Step 3 is unique to the non-OECD survey, where we present the fact that OECD countries adopt supply chain policies. This is a necessary step for two reasons: (i) to provide real-world political context to non-OECD respondents but, more importantly, (ii) to allow subsequent survey components to measure the demand for *aligning* non-OECD policies with OECD first-mover efforts. Finally, step 4 presents the main question of the following experiment to respondents.

Hence, in sum, while the contextual setting of supply chain production and consumption differs, the design and evaluation of the conjoint experimental components remain identical in the OECD and non-OECD surveys. As a result, assessments of preference alignment – more specifically, in the non-OECD survey, support for aligning policies with OECD efforts – towards the 'common goal' of improving production conditions by regulatory policy can be made.

Second, we further examine how acceptability of supply chain policies varies by individual participant characteristics. Drawing upon models of environmental and trade policy preference formation, we expect substantial variation in policy preferences conditional on individual skill and income levels and environmental attitudinal disposition.

First, due to cross-national heterogeneity in educational structures, we use a measure of years of education completed to evaluate participants' skill levels. In the corresponding survey item, we ask participants "In total, how many years were you in mandatory school, college/university, or other school-based training (not including in-company training)?", whereby the response scale ranges from 0 to 20 years of completed education. Additionally, we ask respondents to indicate their household income using country-specific income deciles.

Second, we identify variance in policy support by two environmental measures, pro-environmental attitudes, and environmental impact perceptions. We adapt items from an environmentalism scale implemented by the International Social Survey Programme,[47] First, a construct which captures general 'pro-environmental attitudes', "I do what is right for the environment, even when it costs more money or takes more time". Second, we also identify a single item that directly relates to perceptions of experienced environmental impacts "Environmental problems have a direct effect on my everyday life". Both of these items are evaluated on a Likert-scale ranging from 1 'strongly disagree' to 7 'strongly agree'.

### Resiliency of policy preferences

Lastly, to evaluate the resiliency of citizens' supply chain policy preferences, we adopt an informational vignette experiment, in which participants are randomly assigned to one of five treatment arms directly before the conjoint experiment. Each of these informational vignette treatments highlight a potential consequence of global supply chain policy: domestic improvements in environmental/working conditions, increased exports and job creation, heightened costs (i.e., consumer costs, reduced corporate profits and job loss), and foreign interference in national sovereignty (see Table S4 in Supplemental Information for full informational vignette treatment wording). Participants in the 'control' treatment arm did not receive any additional information before the conjoint experiment.

### Analytical strategy

The conjoint experiment serves as the primary empirical component. We rely on the 'policy support' measure as the dependent variable, which we regress on each of the three policy attributes[31]. In order to enable substantive interpretation of the conjoint analyses, we adopt predicted probabilities[48], calculating the marginal means for each regulatory stringency package[32].

Specifically, we estimate 'policy support' on dummy indicators for each policy attribute level using an ordinary least squares (OLS) model separately for each country. We control for an indicator of informational vignette treatment assignment and estimate robust clustered standard errors to account for correlated errors of multiple responses per participant. We interact the dummy indicators for the policy attribute levels (product-term) to allow for an estimation of a full-factorial model. Each country-level regression model has roughly 20,000 observations – $n = 2000$ respondents by five rounds of conjoint experiment by two policy proposals per round. Then, using these regression estimates, we calculate predicted probabilities of the likelihood to support a given policy proposal for the combinations of attribute levels which correspond to the low-, medium-, and high-stringency policy packages. Accordingly, predicted probabilities can be interpreted as the likelihood of participants within a certain country to support a policy with [low, medium, or high] regulatory stringency in a future national vote. Given that we adopt a subset (three) of the

full-factorial policy attribute cases (36), the analytical sample in this study varies slightly by country: 1761 for Brazil, 1723 for India and 1615 observations for Indonesia. We report analysis of support for the disaggregated policy attribute levels by country in Supplementary Information Figure S1.

Furthermore, to identify how individual characteristics condition supply chain policy support, we adopt a similar methodological approach: We add another parameter of the subgroup characteristic as an interaction term with the policy attribute levels in the OLS regression equations. We then again calculate predicted probabilities at the combination of levels that correspondent to the low-, medium-, and high-stringency packages at substantive levels for each of the subgroup characteristics. For example, we calculate the predicted probability of support for low-stringency policy packages by pro-environmental attitudes at [1 'strongly disagree' to 7 'strongly agree'].

For the estimations of preference resiliency to information provision, we adopt a similar methodology, only in this case, we interact the policy attributes by informational vignette treatment assignment in the OLS regressions. We then again use the OLS estimates to calculate the predicted probability at the combination of levels that correspond to the low-, medium-, and high-stringency package by each vignette treatment condition.

Lastly, we draw on a different set of data (see above) to analyse expectations regarding the benefit and cost consequences associated with varied supply chain policy designs. We assess these perceptions by separately regressing each of the 6 benefit and cost items on a nominal indicator for regulatory stringency policy package treatment assignment separately for each country, using OLS as the estimator. Then, again to facilitate substantive interpretation of these findings, we used the resulting OLS estimates to calculate the marginal mean of each benefit and cost perception by stringency package assignment. Accordingly, these predicted quantities can be interpreted as the expected value of responding to a given benefit or cost statement for participants within a certain country at varying levels of stringency in supply chain policy packages. The number of observations is roughly $n = 2000$ for each country.

All data analyses were performed with Stata SE 16.1.

### Inclusion and ethics
The survey instrument, data collection and storage for this project was approved by the ETH-Zurich's Ethics Committee (EK-2021-N67). The authors declare they have adhered to all ethical regulations for research involving human subjects. Respondents provided their informed consent, and participated voluntarily in the data collection.

### Reporting summary
Further information on research design is available in the Nature Portfolio Reporting Summary linked to this article.

## Data availability
The survey instrument and data are available at Open Science Framework (OSF) with the identifier https://doi.org/10.17605/OSF.IO/YXNW8. The study design was pre-registered at OSF – 10.17605/OSF.IO/7ATUP.

## Code availability
Stata replication code is available along the data at Open Science Framework (OSF) with the identifier https://doi.org/10.17605/OSF.IO/YXNW8.

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

## Acknowledgements

We are grateful to Michael Brander, Gracia Brückmann, Jan Freihardt, Sarah Gomm, Ella Henninger, Matthias Huss, Vally Koubi, Florian Lichtin, David Presberger, Franziska Quoss and Susanne Rhein for valuable feedback on the research design and the survey instrument. Stefano Amberg provided valuable research assistance. We thank Azusa Uji, Do Hyung Shin, Johann Schuur, Maria Murias Munoz, Paul Tromp and Bianca Clément for helping us with the translations of the survey. We are also thankful for the feedback provided at the 2023 Environmental Policy and Governance Conference (EPG) at Glasgow. This research was supported by the Swiss National Science Foundation (SNSF) within the framework of the National Research Programme *Sustainable Economy: resource-friendly, future-oriented, innovative* (NRP 73 grant: 407340–172363, TB) and by SNSF grant no. 182235 *Environmental Burden-Shifting Through International Trade: Driving Forces and Policy Implications, TB*.

## Author contributions

E.K.S., D.K. and T.B. designed the study and contributed to drafting the manuscript. E.K.S. and D.K. led the data collection, E.K.S. analysed and visualised the results, E.K.S. and D.K. led the writing of the manuscript.

## Competing interests

The authors declare no competing interests.
