## [Peer Review File · Nature Communications]

Stringent sustainability regulations for global supply chains are supported across middle-income democraciesREVIEWER COMMENTS

Reviewer #1 (Remarks to the Author):

The present study is an impressive example of public opinion research on the support for sustainability supply chain regulations across diverse high- and middle-income countries. The methods seem sound and were developed with care, and the manuscript is generally well-written. Still, I have a number of recommendations to ensure that the results are understood correctly by readers.

- The way the paper is framed in the introduction initially draws the reader to conclude that what people in exporting countries are evaluating are the supply chain regulations of importing countries; therefore, the high level of support comes at a surprise. It is only when reading the supplementary information that I realized that the question is about a hypothetical supply chain regulation that regulates local firms and their supply chains, mirroring the importing countries' laws. I think this needs to be highlighted better to avoid a false impression, as public opinion on the desirability of importing countries creating such laws may be quite different.

- It would be useful to be more explicit about the types of respondents your data captures or doesn't capture, particularly in producing countries. I assume that the people most affected by the legislation - positively or negatively, for instance smallholder farmers or factory workers - may not be present in the database that is being used. To what extent does this affect your findings?

- I found it curious that the vignettes are used as 'robustness check' rather than to test hypotheses about informational influence themselves (as I've seen it more often). It speaks to the strength of the study that this design was indeed pre-registered; it might be useful to explain these choices in greater depth.

- It would be good to dive deeper into hypotheses or additional data that could explain why the results in Indonesia are so noticeably different from the other countries.

- Smaller notes:

- It would be good to go through and check for clarity of language throughout. Some turns of phrase that were unclear to me, for instance: "How do preferences vary within the Global South and between those in the Global North?" - here it was unclear whether you are comparing countries or segments of populations within countries; "within Indonesia only a narrow majority (50.9%) of supply chain policies are supported" - doesn't this result indicate that only a narrow majority of respondents supported supply chain policies? Or how are you aggregating results here?

- While I understand the need for conciseness, some elements in particular on how the research was conducted should be mentioned earlier, e.g. I would have liked to hear about the policy attributes included in the experimental design (scope, transparency and enforcement capacity) earlier than in the methods section and understand how they were derived.

Reviewer #2 (Remarks to the Author):

Summary of key results. This paper conducts survey research to determine the extent to which each of the general populations of India, of Brazil, and of Indonesia would be accepting or supportive of sustainability regulation within their respective countries to restrict the economic and social burdens created by the manufacturers within their local economies through their upstream participation in global supply chains. Conceptually, this paper defines supply chains to be bilateral, thus distinguishing between upstream production activities concentrated in exporter countries and downstream consumption activities concentrated in importer countries. Generally speaking, upstream production/export activities occur disproportionately in relatively lower-to-middle income countries (referred to as non-OECD economies) while downstream consumption/import activities occur disproportionately in relatively higher-income countries (referred to as OECD economies). The significance of focusing on India, Brazil, and Indonesia is that these countries represent the three largest non-OECD economies. The study's overarching purpose is two-fold: first is to determine each country's propensity to support sustainability regulation within each of their local (production) economies and second is to compare that propensity to the analog propensity of OECD countries to

support sustainability regulation within their local (consumption) economies. The main overall finding is that non-OECD countries (1) are supportive of more rather than less stringent sustainability regulations within their local economies and (2) the support in these OECD countries closely resembles the support for stringency in OECD countries.

Significance, validity, and soundness. Overall, I find the paper to meet requisite standards for consideration. The research question is timely, relevant, interesting, and methodologically rigorous, thus it is poised to make a contribution to the fields of operations and supply chain management, sustainability, and corporate and social responsibility, among others. Moreover, the paper is well written for its clarity, detail, structure, and organization. That all being said, I have a few comments on some of the details with suggestions on how I think the argument could be improved for increased soundness. I present my comments next in increasing order of significance.

Comment 1. Education demographics of sample. In eyeballing Table S5, it is clear that the age and gender demographics of the sample closely track the age and gender demographics of the population it is meant to represent. However, the degree to which the education demographics of the sample track the education demographics of the population is not obvious, especially for Brazil. Given the importance that education level plays in parsing out nuances in the results, I suggest increasing the rigor here, either quantitatively by establishing statistically that the sample distribution indeed is unbiased or qualitatively by addressing any distributional bias in the presentation, interpretation, and discussion of the nuances in the results that depend on the distribution.

Comment 2. Positioning of study relative to literature. As far as I can tell, this study amounts to a follow-up study to a companion study recently published in Nature Sustainability (in 2022) by the same authors. The previously published study is listed as [11] in the current study's references. In the previous study, the main research question was to assess the extent to which the general populations of the countries representing the 12 largest OECD economies are supportive of sustainability regulations within their countries. In contrast, the current study follows a similar research design focusing instead on the 3 largest NON-OECD economies with the intent of providing a comparative analysis of results from the two studies. Given the close relationship between these two studies, I think there would be value-added to the current study if a more explicit framing of the current study's research design were provided in the context of the previous study's findings. I therefore suggest framing the current study as more of a sequel to the previous one by providing a short summary of the previous study (its objective, scope, and findings) at the start of the current study to set the stage and provide useful context for the objective, scope, and findings of the current study.

Comment 3. Framing of results relative to survey design. In light of my previous comment, I have a question about the current study's conclusion presented on pages 18-20 of the current manuscript. As summarized above, the main overall finding of the current study is that there are "high levels of acceptability [of stringent sustainability regulations] in the [three non-OECD] countries studied" with patterns of acceptability in these countries "closely resembling those in OECD countries." However, what's unclear to me from the analysis provided is how much of this conclusion can be stated unconditionally as it is currently stated and how much of it should be conditional on the specific framing of the survey instrument used in the current study. I'll elaborate. In looking closely at Table S1, Step 3 jumps out at me. Step 3 indicates that the survey instrument for the current study of non-OECD economies explicitly introduces the fact that governments of OECD countries (which represent the consumer/importer populations of the global supply chain) are imposing regulations within their borders that would affect by extension the economies of the non-OECD producer/exporter countries. This explicit framing, however, was not used in the previous study focusing on OECD countries. Thus, in effect, the framing of the current study establishes that the regulations being developed in the OECD consumer countries implicitly impose economic sanctions on the non-OECD producer countries by virtue of establishing restrictions on what can and cannot be imported and sold within its consumer-dominated borders. But from the analysis provided in the current study, it's not clear to me if the acceptability of stringent sustainability regulation within the general populations of the non-

OECD countries are anchored by sustainability concerns (as intimated by the study's conclusion) or if it instead is anchored by economic fears of losing the OECD countries as customers. I think this is an important distinction in the context of the study. Accordingly, I think any additional analysis that can convincingly parse out this nuance would be important value-added to the paper's contribution. Alternatively, if additional analysis cannot parse out this nuance, then I think a reframing of the conclusion to anchor it on its conditional nature would be warranted.

Reviewer #3 (Remarks to the Author):

The article aims to investigate the perception of sustainability regulations in developing democracies and uses survey-embedded experiments to gather data. The research is timely and relevant, and the focus on three of the largest democratic non-OECD economies (Brazil, India, and Indonesia) and the 12 largest OECD importers is a significant contribution to the literature. The authors' argument that sustainability regulations are not only accepted but seen as beneficial is an important finding and has potential implications for policy.

However, the article could improve on several fronts:

- 1.Lack of Detailed Methodology: The abstract does not provide enough detail about the nature of the survey-embedded experiments and how they were conducted. For instance, the selection criteria for participants, the design of the survey, and the way the data was processed are not clear.
- 2.Absence of Contextual Analysis: While the high acceptability levels of stringent supply chain regulations across all geographic regions, it does not explore why this might be the case. A comparison of cultural, economic, and political factors across these regions could provide valuable insights.
- 3.Limited Discussion of Cost-Benefit Analysis: The claim that sustainability regulations are perceived as having more benefits than future costs needs further clarification. The article should explore who bears the costs and who reaps the benefits. It would also be interesting to know how this perception might vary between different socioeconomic groups.
- 4.Lack of Evidence for Ideological Priors: The support for stringent regulations is conditioned by individual attitudinal and ideological priors, but does not present clear evidence to support this claim.
- 5.Limited Policy Recommendations: The potential to develop bi- or multilateral sustainability governance frameworks, it does not go into detail on what these frameworks might look like, nor does it provide clear policy recommendations. This would be an essential addition, given the importance of the findings.
- 6.Presentation and Visualization: Please make the presentation of manuscript more crispy and readable.

In conclusion, while the article presents intriguing findings, it could be significantly improved by addressing the above issues. Further detail on the methodology, more contextual analysis, and clearer policy recommendations would substantially enhance the article's contribution to the discourse on sustainability regulations in developing democracies.

Reviewer #4 (Remarks to the Author):

Review of NCOMMS-23-17101 "Stringent sustainability regulations are supported across developing democracies"

Thank you for the opportunity to review the article "Stringent sustainability regulations are supported across developing democracies." The authors seek to explore public preferences for supply chain

management in the three largest non-OECD countries, Brazil, India, and Indonesia, through multiple survey experiments. Given that many previous studies have analyzed supply chain management issues from the developed countries' side, this paper contributes to the literature by providing empirical evidence on the policy preferences of the developing countries' side with original data. Drawing on the results of survey experiments in OECD countries, the authors also assess how people in developing countries express different preferences from those in developed countries. Overall, I commend the authors for providing new insight into the study of supply chain governance. However, I recommend the authors revise the following points before acceptance.

Introduction

1 I would like to see a more elaborate discussion of why and how public preferences in developing countries matter in delivering supply chain management. On page 3, the authors argue "Yet, the effectiveness of such interventions largely depend upon on-the-ground implementation practices throughout the Global South (16), where public acceptability of supply chain regulations could alleviate potential barriers to effectiveness." There is a shared understanding that public opinion matters in policy making process in democratic countries, especially when policy implementation has distributional consequences to the public. However, the government's involvement in supply chain management, including information disclosure targeted in this paper, looks administrative, where there is little room for the public to give a voice. When, why, and how do authors assume public preferences for supply chain management facilitate or hinder its implementation?

2 In the introduction, I encourage the authors to explain more about the motivation for comparing public support for supply chain management between developed countries and developing countries. I think this comparative perspective is a significant part of this paper's contribution. In which way do authors expect public support in developing countries to differ from that in developed countries in the specific context of supply chain management?

Theory and hypotheses

3 In the paper, the authors seek to test propositions with multiple survey experimental settings. However, it is hard to understand which experiment was conducted to test which proposition. First, I suggest the authors formalize four expectations into hypotheses (H1-H4), which can be proposed at the end of each subsection "Macro- and...", "Policy Benefit and...", and "Environmental and...". Thereby, the authors can also integrate the short section "Theoretical Expectations" into the previous section. Then, in "Research Design," I suggest that the authors articulate which experiment was designed to test which hypothesis.

Results

4 The authors present the overview of each experimental setting in "Research Design" and the result of these experiments in the following subsections. However, I find it difficult to follow which experiment provides which result. Alternatively, I suggest that authors discuss experimental design and results in the same subsection. For example, in "High Acceptability of Global Supply Chain Policy Packages...", start with the discussion on the conjoint setting (currently presented on page 8) and then present the conjoint results. Relatedly, for some parts, I struggled to understand the structure of each experimental design before reading the method section. To improve the readability, I recommend the authors add some more sentences excepted from the method part to detail the purpose and procedure of each experiment.

5 Before reading the method section, I have no idea what "Resiliency to Experimental Information Provision" (p15) is about and which experiment this section rests on. This is mainly because any expectation or hypothesis is provided in advance although presented in the method section. I assume that this result builds on the first conjoint experiment combined with the framing experiment. If so, I suggest the author make this point clear in the result section. One possible option is that the authors briefly discuss this robustness check along with the conjoint result in the previous section and move the result (Figure 4) to the Appendix.

Discussion and Conclusion

6 In conclusion, the authors compare public support in OECD and non-OECD countries in parallel. "In particular, the patterns of acceptability in Brazil and India closely resemble those in OECD countries. This overall finding is quite remarkable, given large gaps in economic development between (most) OECD countries and Brazil and India – World Bank average estimates of GDP (PPP) per capita across OECD members are almost three times the estimates for Brazil, and six times the estimates for India. We observe lower levels of acceptability in Indonesia. (P 18)" However, I do not think public support between the two groups is equally comparable because different background information is provided to each group. In the non-OECD countries' version, the introduction of the survey experiment appeals to the potential future economic gains by highlighting that supply chain management policy potentially helps ensure future profits by averting risks ("Soon companies will only be allowed to sell imported products in Europe and North America that meet these increased standards for working conditions and environmental protection."). In the OECD countries' version, the introduction ("Hence, imported products that we buy locally in the United States may be produced abroad...") seems to appeal to the normative considerations rather than the public's economic gains. Overall, people from OECD and non-OECD countries should have evaluated the supply chain management policy based on a different logic, and their results are not equally comparable.

Related to this point, while the introduction of the non-OECD countries' version helps people pay more attention to the benefits of supply chain management, it does not talk about the downside of supply chain management policy, including job or profit losses. Therefore, against the authors' expectation ("To the extent to which regulating global supply chains reduces developing countries' comparative advantage in pollution- or labor-intensive production, ... (p5)"), with the lack of existing knowledge of the costs of supply chain management policy, people may regard it as all "good" and express favorable attitudes. Accordingly, the result "Overall, expectations of supply chain policy benefits exceed cost concerns across all conditions and countries. (p18)" may be partly driven by the wording of the introduction frame.

I encourage the authors to note these points in their discussion/conclusion.

Minor points

7 The first paragraph of "Research Design" that discusses the OECD vs. non-OECD sample (p8) does not seem to fit into this location, and the authors may want to move this part somewhere else.

8 In the presentation of the conjoint result, what 'low,' 'middle,' and 'high' levels of regulatory conditions mean in Panel B (p10-11) is unclear. There should be 36 combinations of policy packages in total. How do the authors categorize them as 'low,' 'middle,' and 'high'?

9 In the previous sections, the authors do not mention the comparison between non-OECD vs OECD countries based on subsample analysis by subgroup characteristics (Panel D in Figure 3). The discussion and conclusion section does not highlight this dimension either. The authors may want to elaborate on expectations/discussion on this point further or move this result (Panel D) to the appendix if the authors regard this analysis as supplemental.

10 In the method section, I encourage the authors to make it clear that subsample analyses by individual characteristics as well as comparisons between non-OECD and OECD countries build on the first conjoint experiment, to avoid confusion. Some, especially those unfamiliar with the survey experiment, may think the authors conducted another experiment.

Author Response Memo - NCOMMS-23-17101

First, we want to thank the editorial team at *Nature Communications* and the four anonymous reviewers for these helpful, thorough and constructive comments. We believe these suggestions allowed us to make significant improvements in our manuscript and have stimulated our thinking on several issues and helped us clarify many arguments. We have also incorporated the changes suggested by the Referees into the revised manuscript.

Please find our response to each comment (listed in italics) listed below, sorted by expert reviewer.

Again, we thank the editor and the expert reviewers for these comments and feedback. We believe that the manuscript has greatly improved as a result.

Reviewer #1

Reviewer Comment 1.1

The present study is an impressive example of public opinion research on the support for sustainability supply chain regulations across diverse high- and middle-income countries. The methods seem sound and were developed with care, and the manuscript is generally well-written. Still, I have a number of recommendations to ensure that the results are understood correctly by readers.

Author Response 1.1

Thank you very much for the detailed comments and constructive feedback on this manuscript.

Reviewer Comment 1.2

The way the paper is framed in the introduction initially draws the reader to conclude that what people in exporting countries are evaluating are the supply chain regulations of importing countries; therefore, the high level of support comes at a surprise. It is only when reading the supplementary information that I realized that the question is about a hypothetical supply chain regulation that regulates local firms and their supply chains, mirroring the importing countries' laws. I think this needs to be highlighted better to avoid a false impression, as public opinion on the desirability of importing countries creating such laws may be quite different.

Author Response 1.2

We thank you very much for this response. We have reviewed the experimental designs for both of these surveys - and agree with these comments. Accordingly, we have revised the manuscript to reflect these substantive differences – where the correct interpretation is in regard to public preferences regarding new domestic laws which mirror those which those being implemented by importing countries. Specifically, we revised and extended the discussion of the similarities and differences between the OECD and non-OECD study designs in the Methods and acknowledge the limitations of direct comparisons more explicitly within the discussion. We further checked and improved text passages throughout the entire paper, which could potentially be confusing regarding what regulatory realm and scope we are studying.

Reviewer Comment 1.3

It would be useful to be more explicit about the types of respondents your data captures or doesn't capture, particularly in producing countries. I assume that the people most affected by the legislation - positively or negatively, for instance smallholder farmers or factory workers - may not be present in the database that is being used. To what extent does this affect your findings?

Author Response 1.3

This is also a well-taken comment. The sampling strategy focuses on developing a nationwide sample of respondents with quotas for age, gender and education. We do not sample based upon employment sector.

We did perform further analyses (please see newly included Figure 2) for policy preferences by household income decile. We did not find strong patterns of differences across income deciles. Overall, there are very minimal differences in support for high and medium policy packages across income deciles in these three countries. There are some smaller differences in support for lower stringency policy packages in Brazil, where higher income respondents are $\sim 7\%$ less likely to support lower stringency packages than lower income respondents. We interpret these smaller effect sized findings more as evidence of some increased demand for increased provisions amongst higher income respondents in Brazil.

Given the developing state of this current literature, the scope of this project was to understand cross-national preference formation towards supply chain policies. This cross-national scope does not directly address some aspects of within-country heterogeneities, such as the differential consequences of being exposed to new regulations. Fitting with the growing literature on distributional impacts of environmental policies and citizen preferences, we strongly agree with the suggestion of follow-up work on these more foundational findings, exploring preference formation amongst such subgroups within country case studies. We have added this call for future research into the discussion). Further, as we do not directly address these distributional impacts in this study design - we more directly acknowledge this limitation in the discussion as well.

Reviewer Comment 1.4

I found it curious that the vignettes are used as 'robustness check' rather than to test hypotheses about informational influence themselves (as I've seen it more often). It speaks to the strength of the study that this design was indeed pre-registered; it might be useful to explain these choices in greater depth.

Author Response 1.4

Thank you for this detailed, constructive comment as well. This approach builds upon previous empirical findings conducted by the authorship group - where we have commonly found that policy preferences formations are often robust against informational provisioning experimental treatments. For example, in the partner paper on this project focusing on supply chain preferences within OECD countries, we also adopted the same approach as a robustness check (see Kolcava et al., 2023, Nature Sustainability, Figure 6).

We also agree that this is a finding that can be further highlighted, as suggested. In this case, we have moved this section to the "main results", and added an expectation for sensitivity to cost informational treatments (Expectation 4)

Reviewer Comment 1.5

It would be good to dive deeper into hypotheses or additional data that could explain why the results in Indonesia are so noticeably different from the other countries.

Author Response 1.5

We agree with the reviewer that the observed gap between Indonesia and India is rather unexpected. However, we interpret this result to be more "surprising" for India than for Indonesia given that, out of the three study countries, India has the lowest income-level in terms of GDP per capita. As noted previously, we conducted exploratory analyses for further subgroups, and did not find substantially different interaction effects between India and Indonesia based upon income. Hence, potentially, the difference may occur due to structural differences such as the composition and complexity of the economy (India's is ranked higher than Indonesia's) or the main export destinations (the US for India, China for Indonesia). Ultimately, we consider systematic inquiry into these drivers paths for future research.

Further, as part of the exploratory analyses of subgroups (Fig S2), we investigated the role of attitudinal characteristics in India. First, we explored the role of global identity, using an item from the ISSP (2016) "I feel more like a citizen of the world than of any country". The substantive effect of global identity is most prominent in India, where those that identify as a citizen of the world are more likely to support policies across all stringency levels. This is particularly notable as India has by far the highest proportion of respondents that have a global identity - where 54% either agree or strongly agree with this statement. For reference, the percentage of global citizens is 34% in Brazil, 33% in Indonesia and 21% across the 12 OECD countries. Similarly, trust in institutions is associated with increased support for policies across stringency levels in India. We again find comparatively higher levels of trust in institutions in India (48% of respondents have high levels of trust). These preliminary analyses suggest that the "surprising" findings within India may be more related to individual attitudes, than to distributional factors. Yet, these exploratory findings would need to be more directly addressed in a follow-up study for confirmation.

Lastly, we have also revisited the full-factorial results of the conjoint experiment (see Figure S2), noting that respondents in Indonesia are similarly likely to support high stringency policy designs as people in Brazil and India if these are applied to larger firms only (e.g. $\geq 25,000$ employees). While, once the regulations are applied to smaller firms (e.g. ≥ 25 employees) support drops by 10-15%. Accordingly, citizens in Indonesia may still support more stringent policy packages, yet have concerns about the effects on smaller-scale producers.

In response to these findings, we revised and extended the interpretation of the differential results for Indonesia and India within the discussion section.

Reviewer Comment 1.6

Smaller notes:

It would be good to go through and check for clarity of language throughout. Some turns of phrase that were unclear to me, for instance: "How do preferences vary within the Global South and between those in the Global North?" - here it was unclear whether you are comparing countries or segments of populations within countries; "within Indonesia only a narrow majority (50.9%) of supply chain policies are supported" - doesn't this result indicate that only a narrow majority of respondents supported supply chain policies? Or how are you aggregating results here?

Author Response 1.6

Thank you, we have further revised the manuscript for clarity as well.

Reviewer Comment 1.7

While I understand the need for conciseness, some elements in particular on how the research was conducted should be mentioned earlier, e.g. I would have liked to hear about the policy attributes included in the experimental design (scope, transparency and enforcement capacity) earlier than in the methods section and understand how they were derived.

Author Response 1.7

We thank you for these comments as well, as it is echoed by other reviewers as well. We have further elaborated on the research design section within the main text to make the research design and empirical approaches clearer to the reader.

Reviewer #2

Reviewer Comment 2.1

Summary of key results. This paper conducts survey research to determine the extent to which each of the general populations of India, of Brazil, and of Indonesia would be accepting or supportive of sustainability regulation within their respective countries to restrict the economic and social burdens created by the manufacturers within their local economies through their upstream participation in global supply chains. Conceptually, this paper defines supply chains to be bilateral, thus distinguishing between upstream production activities concentrated in exporter countries and downstream consumption activities concentrated in importer countries. Generally speaking, upstream production/export activities occur disproportionately in relatively lower-to-middle income countries (referred to as non-OECD economies) while downstream consumption/import activities occur disproportionately in relatively higher-income countries (referred to as OECD economies). The significance of focusing on India, Brazil, and Indonesia is that these countries represent the three largest non-OECD economies. The study's overarching purpose is two-fold: first is to determine each country's propensity to support sustainability regulation within each of their local (production) economies and second is to compare that propensity to the analog propensity of OECD countries to support sustainability regulation within their local (consumption) economies. The main overall finding is that non-OECD countries (1) are supportive of more rather than less stringent sustainability regulations within their local economies and (2) the support in these OECD countries closely resembles the support for stringency in OECD countries.

Significance, validity, and soundness. Overall, I find the paper to meet requisite standards for consideration. The research question is timely, relevant, interesting, and methodologically rigorous, thus it is poised to make a contribution to the fields of operations and supply chain management, sustainability, and corporate and social responsibility, among others. Moreover, the paper is well written for its clarity, detail, structure, and organization. That all being said, I have a few comments on some of the details with suggestions on how I think the argument could be improved for increased soundness. I present my comments next in increasing order of significance.

Author Response 2.1

Thank you for the excellent summary and kind words about the project, as well as for the constructive comments below.

Reviewer Comment 2.2

Reviewer Comment 1. Education demographics of sample. In eyeballing Table S5, it is clear that the age and gender demographics of the sample closely track the age and gender demographics of the population it is meant to represent. However, the degree to which the education demographics of the sample track the education demographics of the population is not obvious, especially for Brazil. Given the importance that education level plays in

parsing out nuances in the results, I suggest increasing the rigor here, either quantitatively by establishing statistically that the sample distribution indeed is unbiased or qualitatively by addressing any distributional bias in the presentation, interpretation, and discussion of the nuances in the results that depend on the distribution.

Author Response 2.2

This point is very well taken, and we appreciate the detailed read into the sample characteristics of these countries. This study recruited respondents from a commercial, online-based panel provider (Dynata), an increasingly common practice in survey data research. The panels are opt-in designs, and are therefore non-proportional sample methodologies. This is a limitation that we have noted within the research design section.

In order to minimise biases in our sample, we adopted quotas in our recruitment for age*gender (interlocked) and education. In our experience with online panels, the most difficult quota to fill is often for individuals with lower education. This proved to particularly be the case for our sample in Brazil, where the panel provider (and their partners) were unable to recruit any lower educated respondents to the survey. In this case, the analytical sample has proportionally fewer respondents with lower education than the population distribution in Brazil. This is a limitation, which we have further added to the discussion to make these differences in sample and population in Brazil clearer to the reader.

We suggest that the potential bias resulting from these sample differences in Brazil is minimal for two primary reasons. First, the overall effect size of educational attainment in Brazil is substantively relatively small (Figure 3, Panel A). Support for high and low stringency policy packages does not vary much across years of education in Brazil. There are some differences for medium stringency packages, but here, we find that lower educated respondents are roughly 8-10% more likely to support medium stringency packages than higher educated responses. These findings suggest that if there is a bias resulting from educational attainment, it would be towards decreased levels of policy support for medium stringency packages - suggesting that our findings of high-levels of support for these designs may be more conservative in Brazil.

Second, to formally test for these differences, we have reanalysed Figures 1, controlling for education and income. Here, we find no differences to the findings presented in the main text (see Figure M1). Accordingly, we suggest that potential bias in our reported estimates resulting from these sampling differences are likely very small.

Reviewer Comment 2.3

Reviewer Comment 2. Positioning of study relative to literature. As far as I can tell, this study amounts to a follow-up study to a companion study recently published in Nature Sustainability (in 2022) by the same authors. The previously published study is listed as [11] in the current study's references. In the previous study, the main research question was to assess the extent to which the general populations of the countries representing the 12 largest OECD economies are supportive of sustainability regulations within their countries.

Figure M1: Support for global supply chain policies in BR, ID, IN, and the 12 largest OECD countries (by imports) controlling for individual education (in years) and household income (in country-specific deciles). Panel A presents the average support across all policy treatment conditions for each region. Panel B displays the marginal means of predicted support for policies by regulatory stringency for each region, with 95% confidence intervals. Both panels use data from choice-experimental survey designs.

In contrast, the current study follows a similar research design, focusing instead on the 3 largest NON-OECD economies with the intent of providing a comparative analysis of results from the two studies. Given the close relationship between these two studies, I think there would be value-added to the current study if a more explicit framing of the current study's research design were provided in the context of the previous study's findings. I therefore suggest framing the current study as more of a sequel to the previous one by providing a short summary of the previous study (its objective, scope, and findings) at the start of the current study to set the stage and provide useful context for the objective, scope, and findings of the current study.

Author Response 2.3

We thank you very much for this response. We have reviewed the experimental designs for both of these surveys and have revised the manuscript to better reflect the commonalities and substantive differences between the two studies. Given that other reviewers encouraged us to elaborate on the differences between the OECD and non-OECD studies, we refrained from framing this study as a pure sequel. We did, however, revised and extend the discussion of the similarities and differences between the OECD and non-OECD study designs in the Methods and further acknowledge the limitations of direct comparisons more explicitly within the discussion. We also checked and improved text passages throughout the entire paper which

could potentially be confusing regarding what regulatory realm and scope we are studying – here, the correct interpretation is in regard to public preferences on new domestic laws in non-OECD countries which mirror those which those being implemented by importing (OECD) countries. As a result, we derive an ‘alignment of preferences’ between OECD and non-OECD countries as our main finding rather than a strict preference congruence.

Reviewer Comment 2.4

Reviewer Comment 3. Framing of results relative to survey design. In light of my previous comment, I have a question about the current study’s conclusion presented on pages 18-20 of the current manuscript. As summarized above, the main overall finding of the current study is that there are “high levels of acceptability [of stringent sustainability regulations] in the [three non-OECD] countries studied” with patterns of acceptability in these countries “closely resembling those in OECD countries.” However, what’s unclear to me from the analysis provided is how much of this conclusion can be stated unconditionally as it is currently stated and how much of it should be conditional on the specific framing of the survey instrument used in the current study. I’ll elaborate. In looking closely at Table S1, Step 3 jumps out at me. Step 3 indicates that the survey instrument for the current study of non-OECD economies explicitly introduces the fact that governments of OECD countries (which represent the consumer/importer populations of the global supply chain) are imposing regulations within their borders that would affect by extension the economies of the non-OECD producer/exporter countries. This explicit framing, however, was not used in the previous study focusing on OECD countries. Thus, in effect, the framing of the current study establishes that the regulations being developed in the OECD consumer countries implicitly impose economic sanctions on the non-OECD producer countries by virtue of establishing restrictions on what can and cannot be imported and sold within its consumer-dominated borders. But from the analysis provided in the current study, it’s not clear to me if the acceptability of stringent sustainability regulation within the general populations of the non-OECD countries are anchored by sustainability concerns (as intimated by the study’s conclusion) or if it instead is anchored by economic fears of losing the OECD countries as customers. I think this is an important distinction in the context of the study. Accordingly, I think any additional analysis that can convincingly parse out this nuance would be important value-added to the paper’s contribution. Alternatively, if additional analysis cannot parse out this nuance, then I think a reframing of the conclusion to anchor it on its conditional nature would be warranted.

Author Response 2.4

We thank the reviewer for this important comment on the relationship between preferences in OECD and non-OECD countries, and in particular, the nuances of how we introduced respondents in both surveys to the conjoint experiment. These are also comments that are similarly shared by Reviewer 4, which we have responded to in Response 4.7 as well.

As a response to these comments, we reviewed the survey instruments and then, as an authorship team, discussed the implications of the wording for the interpretation of our findings. In response, we have substantially edited the framework of the paper, notably

revising the motivation within the introduction and interpretation of the results within the discussion. Moreover, we have extended the methods section to be more explicit about the purpose of each of the steps of the experiment introduction. We thank the reviewer for these comments, which has incited these necessary clarifications and revisions.

Specifically, these revisions have shifted the focus away from ‘hard’ comparisons between public opinion in OECD and non-OECD states, which, as the reviewer rightfully observed, may have been thinly stretched in terms of inferences on the congruence of preference formation processes. Instead, we adjusted the focus of the paper towards the **alignment** of preferences to regulate global supply chains, which we argue, our identical experiment at both the ‘consumer’ and the ‘producer’ ends can identify. That is, the revised interpretation is whether citizens in non-OECD countries are supportive of domestic policies which are in alignment with the extraterritorial measures being taken by OECD ‘consumer’ states.

To that end, Step 3 of the introduction text to the experimental design is highly valuable, as it provides important real-world context to non-OECD respondents, allowing for the subsequent experiment to tap into public support for an ‘alignment’ of domestic policies along OECD members’ moves.

Lastly, whether the wording of step 3 may have affected non-OECD participants’ responses in the conjoint, we presented random respondents with direct treatments on consequences (that is, i) economic costs, ii) improvements in production conditions, iii) national sovereignty shifts and iv) economic gains) of aligning domestic policies along OECD regulatory advances. We do not observe significant effects (Figure 4) of providing this information to respondents directly before the first choice task (that is, after respondents read Step 3 of the introduction). If respondents’ opinions were sensitive towards concerns of economic gains information, we should have observed stronger shifts as a result of the direct experimental vignettes. In this case, we do not believe that Step 3 of the experimental design introduction text has a substantial effect on the observed preferences in non-OECD countries, as these are largely robust against cost and benefit information treatments.

Reviewer #3

Reviewer Comment 3.1

The article aims to investigate the perception of sustainability regulations in developing democracies and uses survey-embedded experiments to gather data. The research is timely and relevant, and the focus on three of the largest democratic non-OECD economies (Brazil, India, and Indonesia) and the 12 largest OECD importers is a significant contribution to the literature. The authors' argument that sustainability regulations are not only accepted but seen as beneficial is an important finding and has potential implications for policy.

However, the article could improve on several fronts:

Author Response 3.1

Thank you for the considered and detailed read of this manuscript, and for the constructive comments.

Reviewer Comment 3.2

1.Lack of Detailed Methodology: The abstract does not provide enough detail about the nature of the survey-embedded experiments and how they were conducted. For instance, the selection criteria for participants, the design of the survey, and the way the data was processed are not clear.

Author Response 3.2

We thank you for this comment, as it is similar shared by other reviewers as well. We have expanded upon the Research Design section within the main text, to make expand upon the sampling, design and analytical approach developed within these analyses.

Reviewer Comment 3.3

2.Absence of Contextual Analysis: While the high acceptability levels of stringent supply chain regulations across all geographic regions, it does not explore why this might be the case. A comparison of cultural, economic, and political factors across these regions could provide valuable insights.

Author Response 3.3

This is also a helpful comment, reflecting concerns shared by Reviewer 4. In order to further contextualise these case studies, we have included a new Table 1 in the main text, which compares the structural characteristics of case study countries.

Reviewer Comment 3.4

3.Limited Discussion of Cost-Benefit Analysis: The claim that sustainability regulations are perceived as having more benefits than future costs needs further clarification. The article should explore who bears the costs and who reaps the benefits. It would also be interesting to know how this perception might vary between different socioeconomic groups.

Author Response 3.4

Thank you again for this comment. Our understanding of this comment is that it raises two larger concerns.

First, when we report the cost-benefit analyses within this paper, we primarily rely upon preference formation follow-up probes (see Figure 2). After the conjoint experiment, we randomly assigned either low, medium or high policy stringency packages to respondents, and tasked them with evaluating the impact of such a proposal across a number of constructs of potential benefits (information, production practices and job creation), as well as by potential costs (to firms, to consumers, to the sovereignty of the country).

These suggest preference formation probes can be analysed across three axes. First, do perceptions of benefits and costs vary by policy stringency (we find that higher stringency packages are perceived as having greater benefits, while there are little differences with regard to cost perceptions to firms and consumers across stringency levels). Second, how do perceptions vary by country (we find little differences for benefit perceptions, especially at higher stringency packages, but for costs, we find that respondents consistently observe comparatively higher cost implications for firms and consumers in India, then Indonesia, and least in Brazil). Third, and most relevant to the comment raised here, we can compare the substantive effect of benefit and cost perceptions using the marginal means. The perceptions are evaluated on a scale of 1 'strongly disagree' to 7 'strongly agree'. We find that across all countries and policy stringency levels, the marginal means for benefits are greater than the marginal means for cost perceptions. For example, for high stringency packages, the marginal means of the three benefit items are all above 5.5 for each country - while the perceptions of costs for firms and individuals are around 5.0 for India, 4.5 for Indonesia and 4.0 for Brazil (roughly). Given these substantive differences, we interpret these findings to suggest that respondents are likely to perceive comparatively greater impacts of the benefits of these proposed policies than cost implications. In response to these comments, we have further revised the methodological approach to this section, as well as the interpretation of these findings, to make these analyses more clear.

Second, this comment raises concerns about differential impacts across population subgroups, a concern which was similarly raised by Reviewer 1 (Comment 1.3). As we note in our response to Reviewer 1, we agree that such distributional differences are of high interest, and should be very much explored in further detail. In this case, we have now also included income as a sub-group condition in our revised analyses (see Revised Figure 3). Notably, we do find substantial differentiation based upon education in India and Indonesia (but notably not in Brazil). But, we do not find strong patterns of differences across income deciles (please see Response 1.3 for further discussion).

We also suggest that such further research is outside the scope of this particular research project. To our knowledge, this is the first comparative survey of supply chain policy preferences between OECD and non-OECD states. In this case, the scope of the project is largely to establish a foundation of comparative analyses - focusing on elements of how preferences vary across these countries, how respondents weigh costs and benefits, and how preferences vary by subgroups within each country. Our hope is that these analyses can provide the groundwork, from which further study can explore the differential pathways of supply chain policy preference formation within each of these countries. Much of this work should attend to identifying who "bears the costs", and "reaps the benefits".

In sum, we very much agree that this is an important avenue of future research (and have noted so in the discussion), and that not further focusing on subgroups that may directly benefit or lose from these policies is a limitation (also noted in the discussion), but we believe that such future work should be completed within subsequent studies, and is outside the scope of this current project.

Reviewer Comment 3.5

4.Lack of Evidence for Ideological Priors: The support for stringent regulations is conditioned by individual attitudinal and ideological priors, but does not present clear evidence to support this claim.

Author Response 3.5

Thank you for this comment. We have further discussed the role of ideological priors as a research team, and agree that there are potential issues including these analyses within this study. Notably, the measurement of ideology varies substantially across country context (particularly outside of western, educated, industrialised, rich democracies). Furthermore, as this comment notes, such dynamics are not well-developed theoretically. In this case, we have chosen to rather report these findings in SI (which could be utilised as a foundation for further researchers), and not motivate, present or discuss the role of ideology within the main text.

Reviewer Comment 3.6

5.Limited Policy Recommendations: The potential to develop bi- or multilateral sustainability governance frameworks, it does not go into detail on what these frameworks might look like, nor does it provide clear policy recommendations. This would be an essential addition, given the importance of the findings.

Author Response 3.6

This feedback is well taken, and is further emphasized by editor and reviewer comments. We also thank the reviewer for pushing us to better frame the relevance of these findings. Accordingly, we have substantially revised the discussion to more clearly outline the policy impacts of these findings.

Reviewer Comment 3.6

6.Presentation and Visualization: Please make the presentation of manuscript more crispy and readable.

Author Response 3.6

We have further revised the text for clarity, and revised the figures to facilitate reader interpretation.

Reviewer Comment 3.7

In conclusion, while the article presents intriguing findings, it could be significantly improved by addressing the above issues. Further detail on the methodology, more contextual analysis, and clearer policy recommendations would substantially enhance the article's contribution to the discourse on sustainability regulations in developing democracies.

Author Response 3.7

Thank you very much for these comments and for this feedback.

Reviewer #4

Reviewer Comment 4.1

Thank you for the opportunity to review the article “Stringent sustainability regulations are supported across developing democracies.” The authors seek to explore public preferences for supply chain management in the three largest non-OECD countries, Brazil, India, and Indonesia, through multiple survey experiments. Given that many previous studies have analyzed supply chain management issues from the developed countries’ side, this paper contributes to the literature by providing empirical evidence on the policy preferences of the developing countries’ side with original data. Drawing on the results of survey experiments in OECD countries, the authors also assess how people in developing countries express different preferences from those in developed countries. Overall, I commend the authors for providing new insight into the study of supply chain governance. However, I recommend the authors revise the following points before acceptance.

Author Response 4.1

Thank you as well for the careful read of this manuscript and for the constructive comments provided.

Reviewer Comment 4.2

Introduction #1 I would like to see a more elaborate discussion of why and how public preferences in developing countries matter in delivering supply chain management. On page 3, the authors argue “Yet, the effectiveness of such interventions largely depend upon on-the-ground implementation practices throughout the Global South (16), where public acceptability of supply chain regulations could alleviate potential barriers to effectiveness.” There is a shared understanding that public opinion matters in policymaking process in democratic countries, especially when policy implementation has distributional consequences to the public. However, the government’s involvement in supply chain management, including information disclosure targeted in this paper, looks administrative, where there is little room for the public to give a voice. When, why, and how do authors assume public preferences for supply chain management facilitate or hinder its implementation?

Author Response 4.2

The reviewer rightfully raises this important point, and we agree we should outline the role of public preferences in policy adoption and implementation in more detail. In response, we have revised the corresponding arguments to the introduction to focus rather on whether citizens in non-OECD states would their governments to implement domestic policies that are in alignment with the extraterritorial measures currently being adopted across OECD states. These arguments focus more around the role of preferences in the policy development/adoption stage, particularly whether citizens prefer diffusion of supply chain measures from OECD into non-OECD domestic policies. In this case, the argumentation is shifted

away from implementation, as suggested, and rather towards the role of policy preferences in designing and adopting policy measures.

Reviewer Comment 4.3

#2 In the introduction, I encourage the authors to explain more about the motivation for comparing public support for supply chain management between developed countries and developing countries. I think this comparative perspective is a significant part of this paper's contribution. In which way do authors expect public support in developing countries to differ from that in developed countries in the specific context of supply chain management?

Author Response 4.3

Similar to our response above, we appreciate the encouragement to extend the motivation of our comparative study. In response, we have elaborated on why we would study the alignment of preferences within developed and developing countries. This is a concern which was also raised by Reviewer 2, which we have responded to more extensively in our Responses 2.3 and 2.4.

Reviewer Comment 4.4

Theory and hypotheses

#3 In the paper, the authors seek to test propositions with multiple survey experimental settings. However, it is hard to understand which experiment was conducted to test which proposition. First, I suggest the authors formalize four expectations into hypotheses (H1-H4), which can be proposed at the end of each subsection "Macro- and...", "Policy Benefit and...", and "Environmental and...". Thereby, the authors can also integrate the short section "Theoretical Expectations" into the previous section. Then, in "Research Design," I suggest that the authors articulate which experiment was designed to test which hypothesis.

Author Response 4.4

Thank you, we fully agree with this comment. Given the more exploratory nature of these analyses (to our knowledge, this is the first analysis of citizen preferences to supply chain policies in non-OECD countries), we do not develop hypotheses (as this language is a bit stronger). Rather, we have revised the motivation and theory section to include a set of 'expectations'. The hope is that this will give further structure to the research design and analytical approach that follows. In this regard, we have also added direct references to these expectations in the results and discussion of the paper, to make this structure clearer to the reader.

Reviewer Comment 4.5

Results #4 The authors present the overview of each experimental setting in "Research Design" and the result of these experiments in the following subsections. However, I find

it difficult to follow which experiment provides which result. Alternatively, I suggest that authors discuss experimental design and results in the same subsection. For example, in “High Acceptability of Global Supply Chain Policy Packages...,” start with the discussion on the conjoint setting (currently presented on page 8) and then present the conjoint results. Relatedly, for some parts, I struggled to understand the structure of each experimental design before reading the method section. To improve the readability, I recommend the authors add some more sentences excerpted from the method part to detail the purpose and procedure of each experiment.

Author Response 4.5

Thank you for these comment, as these are also concerns shared by other reviewers as well. We have revised the Research Design section for clarity, and have connected these experimental designs more directly to the expectations developed in the motivation and theory section, to help better structure the argument and make these analytical approaches clearer to the reader.

Reviewer Comment 4.6

#5 Before reading the method section, I have no idea what “Resiliency to Experimental Information Provision” (p15) is about and which experiment this section rests on. This is mainly because any expectation or hypothesis is provided in advance although presented in the method section. I assume that this result builds on the first conjoint experiment combined with the framing experiment. If so, I suggest the author make this point clear in the result section. One possible option is that the authors briefly discuss this robustness check along with the conjoint result in the previous section and move the result (Figure 4) to the Appendix.

Author Response 4.6

Thank you for this comment. This section was also highlighted as being unclear by other reviewers. In this case, we have revised the in-text methods explanation, and added a separate section, with the goal of clarifying these analyses.

Reviewer Comment 4.7

Discussion and Conclusion

#6 In conclusion, the authors compare public support in OECD and non-OECD countries in parallel. “In particular, the patterns of acceptability in Brazil and India closely resemble those in OECD countries. This overall finding is quite remarkable, given large gaps in economic development between (most) OECD countries and Brazil and India – World Bank average estimates of GDP (PPP) per capita across OECD members are almost three times the estimates for Brazil, and six times the estimates for India. We observe lower levels of acceptability in Indonesia. (P 18)” However, I do not think public support between the two groups is equally comparable because different background information is provided to

each group. In the non-OECD countries' version, the introduction of the survey experiment appeals to the potential future economic gains by highlighting that supply chain management policy potentially helps ensure future profits by averting risks ("Soon companies will only be allowed to sell imported products in Europe and North America that meet these increased standards for working conditions and environmental protection."). In the OECD countries' version, the introduction ("Hence, imported products that we buy locally in the United States may be produced abroad...") seems to appeal to the normative considerations rather than the public's economic gains. Overall, people from OECD and non-OECD countries should have evaluated the supply chain management policy based on a different logic, and their results are not equally comparable.

Related to this point, while the introduction of the non-OECD countries' version helps people pay more attention to the benefits of supply chain management, it does not talk about the downside of supply chain management policy, including job or profit losses. Therefore, against the authors' expectation ("To the extent to which regulating global supply chains reduces developing countries' comparative advantage in pollution- or labor-intensive production, ... (p5)"), with the lack of existing knowledge of the costs of supply chain management policy, people may regard it as all "good" and express favorable attitudes. Accordingly, the result "Overall, expectations of supply chain policy benefits exceed cost concerns across all conditions and countries. (p18)" may be partly driven by the wording of the introduction frame. "

I encourage the authors to note these points in their discussion/conclusion.

Author Response 4.7

We thank you very much for this response. We have thoroughly reviewed and discussed these comments as an authorship team, and from our read, these comments raise two larger issues.

First, the comparability of preferences between OECD and non-OECD countries. These are concerns that were similarly raised by Reviewer 2, Comment and Response 2.4 (as noted above). As we note in this response, we have shifted the logic away from direct, absolute comparisons - because as these comments note, the non-OECD respondents are not evaluating the actions of OECD countries, but rather their preferences for whether they want their country's domestic policies to be brought into alignment. In this case, we have revised the motivation and discussion of the results to emphasize the role of preferences towards alignment.

Second, this comment raises concerns about differences in the introduction wording for the experimental designs (notably the differences in Steps 3 and 4). In particular, this comment raises concerns that this framing does not present the policy in a way that directly addresses potential losses, including job or profit losses. We agree with the comment that the conjoint introduction (particularly Step 3) does not include a statement about potential losses. But, we believe that such a statement is first necessary for the experimental design - as it provides 'real-world' context to why a non-OECD country would want to implement such domestic-based measures which are in alignment with the extraterritorial measures taken across the

OECD. If this Step 3 is not included, we are unable to perform the analyses of preferences towards regulatory alignment.

Further, we also believe that these statements in the conjoint introduction do not shape policy preferences. Here we rely upon the findings from the experimental vignette treatments (which are presented to the respondent after the conjoint introduction, directly before they are directed to the choice experiment). We randomly assign respondents to 1 of 5 different treatment allocations (see Table S3). 2 of these treatment allocations are for statements which emphasize the benefits of these new measures (improved conditions, increased trade and jobs), 2 of these treatment allocations are towards potential costs/losses (consumer costs and job loss, threats to sovereignty) and the final treatment allocation is to a control, where no information is presented (e.g. the respondents only have seen the conjoint introduction text). We find that respondent policy preferences in non-OECD countries are robust against all of these forms of informational provisioning (see Figure 4). There are no significant (or substantive) differences in the evaluation of low, medium or high stringency packages based upon these 5 treatment allocations. In sum, we suggest that Step 3 of the conjoint introduction is necessary, as it provides the required background for the policy evaluation, and in the end, we do not find evidence that this wording has shifted policy preferences (as these are robust against multiple different forms of informational treatments).

Reviewer Comment 4.8

Minor points

#7 The first paragraph of “Research Design” that discusses the OECD vs. non-OECD sample (p8) does not seem to fit into this location, and the authors may want to move this part somewhere else.

Author Response 4.8

Thank you. We have reformatted this part of the manuscript to include a new section (Research Design). Within this Research Design section, we have separated these out into a “Case Studies” subsection where we present structural differences between the 3 non-OECD countries studies here, and into an “Analytical Strategy” subsection, where we have revised and clarified our study design and approach.

Reviewer Comment 4.9

#8 In the presentation of the conjoint result, what ‘low,’ ‘middle,’ and ‘high’ levels of regulatory conditions mean in Panel B (p10-11) is unclear. There should be 36 combinations of policy packages in total. How do the authors categorize them as ‘low,’ ‘middle,’ and ‘high’?

Author Response 4.9

Thank you for this feedback as well. We have revised the analytical strategy to make our approach more clear. Here, we make direct reference to the further description of the

stringency packages in the Methods section, including Table 2.

Further, as this comment notes, there is a full-factorial of 36 potential combinations. For parsimony, here we only present 3 of these combinations (as idealized case studies of low, medium and high policy packages). Yet, the full factorial also provides important information. Accordingly, we have added in a new figure to SI (see Figure ??), displaying this information.

Reviewer Comment 4.10

#9 In the previous sections, the authors do not mention the comparison between non-OECD vs OECD countries based on subsample analysis by subgroup characteristics (Panel D in Figure 3). The discussion and conclusion section does not highlight this dimension either. The authors may want to elaborate on expectations/discussion on this point further or move this result (Panel D) to the appendix if the authors regard this analysis as supplemental.

Author Response 4.10

We agree with this comment. Our revised approach is to first focus on whether the preferences of non-OECD citizens are aligned with those from OECD countries (Figure 1), and then to explore the preference formation of citizens in non-OECD countries in Figure 2 and 3. In this case, the only place where data from OECD countries is presented is now in Figure 1, to explore these questions regarding alignment.

Reviewer Comment 4.11

#10 In the method section, I encourage the authors to make it clear that subsample analyses by individual characteristics as well as comparisons between non-OECD and OECD countries build on the first conjoint experiment, to avoid confusion. Some, especially those unfamiliar with the survey experiment, may think the authors conducted another experiment.

Author Response 4.11

Thank you for this feedback, we can also see how this could be confusing. We have revised the analytical strategy to hopefully make these approaches more clear.

REVIEWER COMMENTS

Reviewer #1 (Remarks to the Author):

I thank the authors for a comprehensive revision and their thoughtful engagement with my previous comments. I feel that the previous comments, both by myself as well as those of other reviewers, were sufficiently addressed and support the publication of this manuscript, subject to the following minor framing issues being addressed:

- In the new title and abstract, you now speak of "sustainability standards". I would move away from this reframing and back toward the original "regulation" or "supply chain policy" framing, given that there is a very large literature on sustainability standards, most of which are voluntary and third-party or multi-stakeholder schemes. I don't think using this language does the paper any favors in terms of signaling its policy relevance and novelty, and may be confusing for readers.

- I would also be more consistent in whether you are talking about middle-income, 'lower and middle-income', or 'low and middle-income' countries. There is a variety of these terms in the paper. Both for accuracy and generalizability reasons, given that you are analyzing three middle-income countries, I would probably limit myself to also constructing expectations and creating implications for that group of countries (which is still significant).

- Finally, I would advise one last close read of the manuscript to eliminate some remaining typos (e.g. "Support suggestions that citizen of within developing countries", "supply chains policy" - I would call it "supply chain policy")

P.S. I will note that the submitted clean manuscript seems to have been the original version and that I was only working off the version with track changes, which was quite hard to read.

Reviewer #2 (Remarks to the Author):

Referee report on Nature Communications manuscript NCOMMS-23-17101A "Stringent sustainability regulations are supported across developing democracies" (revised)

This is my second time reviewing this paper. In the previous round I made three general comments in increasing order of significance that I thought needed to be addressed. In my reading of the revision and the response provided to my previous comments, I think the first two of those comments have now been addressed.

Regarding my third comment, while I very much appreciate the conscientious reply provided in the response document, I nevertheless remain concerned with how the revised paper treats the issue.

To highlight my remaining concern, I will point to the "Materials and Methods" section of the manuscript. There, on Page 28, in describing the survey instruments for participants from non-OECD vs. OECD democratic economies, it states: "Accordingly, the introductory information leading into the conjoint experiment was different (Table S1) for the non-OECD and OECD contexts." In my mind, the difference that is referenced on Page 28 is fundamental and, as a result, needs to be underscored and leveraged in the body of the paper to anchor the expectations and results from the study. Currently, that anchoring is missing.

While there are contextual differences between the introductory information provided to OECD participants versus non-OECD participants, the fundamental difference as I see it boils down to the following statement that is provided to non-OECD participants but not to OECD participants (stress added):

Many governments in Europe and North America are currently adopting new laws for companies that

import goods into their country. These laws would require companies to disclose more information about the working and environmental conditions under which imported products are made overseas in countries like [yours]. **Soon companies will only be allowed to sell imported products in Europe and North America that meets these increased standards for working conditions and environmental protection.**

I can appreciate that the survey instrument is identical for non-OECD participants and OECD participants alike. However, I think this different framing of the survey instrument for different participants should have material effect on the meaning of the results. To be specific, I will highlight two places in the manuscript in particular.

First, on Page 8, in summarizing what theory would suggest should be expected from the results of the survey, the manuscript indicates that “First, we expect that, relative to [OECD] countries, we will find less public support for stringent sustainability regulation of global supply chains in [non-OECD] countries.” But, in light of the disclaimer provided to non-OECD survey participants highlighted above, I would say that the survey is not designed to answer one way or another whether non-OECD countries have more or less public support for stringent vs. non-stringent standards. Rather, the survey is designed to answer whether non-OECD countries have more or less public support for standards that are aligned vs. misaligned with the requirements of OECD countries.

Second, on Page 11, in summarizing the baseline results of the survey, the manuscript indicates: “In sum, these findings suggest broad acceptability of stringent, sustainability-oriented global supply chain policies across Brazil, India, and OECD.” But, in light of the disclaimer highlighted above, I would argue that this is not what the results suggest. Rather, I think the results suggest something more nuanced. While I agree that the results do, indeed, suggest broad acceptability of stringent standards for OECD countries, I think for non-OECD countries the results suggest a propositional statement more along the following lines:

If OECD countries adopt stringent sustainability standards for the products they import, then there would be broad acceptability of stringent, sustainability-oriented global supply chain policies across Brazil and India.

For what it’s worth, I think one naïve explanation for these results could be the following: If OECD countries restrict imports in this way, then the peoples of Brazil and India are against leaving it to the manufacturers in their countries to meet the standards on their own and would rather have the government intervene with regulations to ensure that the manufacturers comply with the standards.

In summary, I think the paper would be stronger if it built its argument from start to finish on the nuanced differences highlighted in this report.

Reviewer #3 (Remarks to the Author):

Thank you for submitting the revised version of manuscript. However, the revised version looks very promising and significantly improved, the authors should focus on literature section. There are some recent and relevant studies are missing. The authors should do some efforts to revise the literature section. Moreover, some of sustainability standards e.g ISO 9001, ISO 14001, SA 8000, ISO 26000, and GRI related studies should be included in the literature. Please align your study objectives with the updated literature contribution.

Reviewer #4 (Remarks to the Author):

Thank you. My concerns are addressed by the authors in the revised manuscript.

Author Response Memo — NCOMMS-23-17101B

First, we want to thank the editorial team at *Nature Communications* and the four anonymous reviewers for the second round of these helpful, thorough and constructive comments. We believe these suggestions allowed us to make further improvements to our manuscript. We have also incorporated the changes suggested by the Referees into the revised manuscript.

We wish to bring to the reviewers' attention that an error has occurred as the original manuscript was re-sent out for revision. Reviewer 1 helpfully noted this issue, and was able to provide their comments using the tracked changes version (albeit with some understandable difficulty). We believe this may have caused some confusion for other reviewers as well. We greatly apologise for any resulting inconveniences.

We have responded to each of the reviewer comments below (listed in italics), sorted by expert reviewer.

Again, thank you very much for the constructive comments and feedback.

Reviewer #1

Reviewer Comment 1.1

I thank the authors for a comprehensive revision and their thoughtful engagement with my previous comments. I feel that the previous comments, both by myself as well as those of other reviewers, were sufficiently addressed and support the publication of this manuscript, subject to the following minor framing issues being addressed:

Author Response 1.1

Thank you again for the constructive comments as well.

Reviewer Comment 1.2

In the new title and abstract, you now speak of "sustainability standards". I would move away from this reframing and back toward the original "regulation" or "supply chain policy" framing, given that there is a very large literature on sustainability standards, most of which are voluntary and third-party or multi-stakeholder schemes. I don't think using this language does the paper any favors in terms of signaling its policy relevance and novelty, and may be confusing for readers.

Author Response 1.2

We thank you for this feedback, as this point is well taken. The analysis focuses on the regulations of global supply chains, and not the development of voluntary or private firm-initiated schemes. In this case, we have removed this language as suggested.

Reviewer Comment 1.3

I would also be more consistent in whether you are talking about middle-income, 'lower and middle-income', or 'low and middle-income' countries. There is a variety of these terms in the paper. Both for accuracy and generalizability reasons, given that you are analyzing three middle-income countries, I would probably limit myself to also constructing expectations and creating implications for that group of countries (which is still significant).

Author Response 1.3

We thank you for noting these inconsistencies and for this suggestion. We agree that 'middle income' best describes these three countries. We have revised the manuscript in this regard.

Reviewer Comment 1.4

Finally, I would advise one last close read of the manuscript to eliminate some remaining typos (e.g. "Support suggestions that citizen of within developing countries", "supply chains

policy” - I would call it ”supply chain policy”)

Author Response 1.4

We thank you for noting these typographical errors, and have further revised the manuscript for copy editing.

Reviewer Comment 1.5

P.S. I will note that the submitted clean manuscript seems to have been the original version and that I was only working off the version with track changes, which was quite hard to read.

Author Response 1.5

Thank you very much for alerting us to this issue. It appears there has been an error somewhere along the line, and we greatly apologise for this inconvenience and for any frustration this may have caused you and the fellow reviewers.

Reviewer #2

Reviewer Comment 2.1

This is my second time reviewing this paper. In the previous round I made three general comments in increasing order of significance that I thought needed to be addressed. In my reading of the revision and the response provided to my previous comments, I think the first two of those comments have now been addressed.

Regarding my third comment, while I very much appreciate the conscientious reply provided in the response document, I nevertheless remain concerned with how the revised paper treats the issue.

Author Response 2.1

Thank you very much for your constructive comments and continued feedback on this manuscript.

Reviewer Comment 2.2

To highlight my remaining concern, I will point to the “Materials and Methods” section of the manuscript. There, on Page 28, in describing the survey instruments for participants from non-OECD vs. OECD democratic economies, it states: “Accordingly, the introductory information leading into the conjoint experiment was different (Table S1) for the non-OECD and OECD contexts.” In my mind, the difference that is referenced on Page 28 is fundamental and, as a result, needs to be underscored and leveraged in the body of the paper to anchor the expectations and results from the study. Currently, that anchoring is missing.

While there are contextual differences between the introductory information provided to OECD participants versus non-OECD participants, the fundamental difference as I see it boils down to the following statement that is provided to non-OECD participants but not to OECD participants (stress added):

*Many governments in Europe and North America are currently adopting new laws for companies that import goods into their country. These laws would require companies to disclose more information about the working and environmental conditions under which imported products are made overseas in countries like [yours]. **Soon companies will only be allowed to sell imported products in Europe and North America that meets these increased standards for working conditions and environmental protection.***

I can appreciate that the survey instrument is identical for non-OECD participants and OECD participants alike. However, I think this different framing of the survey instrument for different participants should have material effect on the meaning of the results. To be specific, I will highlight two places in the manuscript in particular.

First, on Page 8, in summarizing what theory would suggest should be expected from the results of the survey, the manuscript indicates that “First, we expect that, relative to [OECD]

countries, we will find less public support for stringent sustainability regulation of global supply chains in [non-OECD] countries.” But, in light of the disclaimer provided to non-OECD survey participants highlighted above, I would say that the survey is not designed to answer one way or another whether non-OECD countries have more or less public support for stringent vs. non-stringent standards. Rather, the survey is designed to answer whether non-OECD countries have more or less public support for standards that are aligned vs. misaligned with the requirements of OECD countries.

Author Response 2.2

Foremost, we again apologise for any confusion caused by the original manuscript being re-sent out alongside the track changed version (please see our initial comments to all reviewers above). We believe that this may have caused some confusion, as the pages referenced above are from the original version of the manuscript.

Hence, we agree with the substance of this feedback, and these points are well taken. In the first round of revisions, we have indeed substantially modified the manuscript to reflect reviewer suggestions to focus on preferences towards the **alignment** of policy with OECD countries’ rules (rather than an absolute comparison between high-, and middle-income countries). And, as this comment notes, there remain some important areas where these concerns have not been fully addressed (such as expectation 1 mentioned in this comment). Therefore, we have again further revised these sections – including Expectation 1 – to make this argumentation and framing more clear.

Reviewer Comment 2.3

Second, on Page 11, in summarizing the baseline results of the survey, the manuscript indicates: “In sum, these findings suggest broad acceptability of stringent, sustainability-oriented global supply chain policies across Brazil, India, and OECD.” But, in light of the disclaimer highlighted above, I would argue that this is not what the results suggest. Rather, I think the results suggest something more nuanced. While I agree that the results do, indeed, suggest broad acceptability of stringent standards for OECD countries, I think for non-OECD countries the results suggest a propositional statement more along the following lines:

If OECD countries adopt stringent sustainability standards for the products they import, then there would be broad acceptability of stringent, sustainability-oriented global supply chain policies across Brazil and India.

For what it’s worth, I think one naïve explanation for these results could be the following: If OECD countries restrict imports in this way, then the peoples of Brazil and India are against leaving it to the manufacturers in their countries to meet the standards on their own and would rather have the government intervene with regulations to ensure that the manufacturers comply with the standards.

Author Response 2.3

Thank you again for this comment. We agree, this summary interpretation (now on page 13) was not properly reflective of this framing of support for domestic policies in alignment with OECD-implemented regulations. We have revised this section, drawing upon these constructive comments:

”In sum, these findings suggest that in response to increased global supply chain sustainability regulations in the OECD, there is broad acceptability for stringent, sustainability-oriented global supply chain policies within Brazil and India. Furthermore, while the supply chain policies are comparatively less supported in Indonesia, a plurality of citizens are accepting of medium- and high-stringency policy packages.” (p.13)

In addition, we have included further explanation of these results in the discussion as well. We believe that these explanations are supportive of the trading-up arguments noted in the policy implications section:

”First, we find that citizens of non-OECD countries are supportive of domestic policies which align with emerging extraterritorial supply chain measures developed across OECD states, particularly at higher levels of policy stringency. The overall levels of support varies cross-nationally: Our findings in Brazil and India suggest substantial public support for aligning domestic policies with current regulatory shifts within OECD countries. While policy preferences towards alignment are comparatively lower in Indonesia, a majority of respondents support medium- and high-stringency policy designs. Accordingly, these findings suggest that if OECD countries place increased sustainability requirements on imported products, citizens within Brazil and India are supportive of domestic government regulation of production conditions to ensure compliance.” (p.20)

Reviewer Comment 2.4

In summary, I think the paper would be stronger if it built its argument from start to finish on the nuanced differences highlighted in this report.

Author Response 2.4

We again thank you for the constructive comments and thorough suggestions on this manuscript.

Reviewer #3

Reviewer Comment 3.1

Thank you for submitting the revised version of manuscript. However, the revised version looks very promising and significantly improved, the authors should focus on literature section. There are some recent and relevant studies are missing. The authors should do some efforts to revise the literature section. Moreover, some of sustainability standards e.g ISO 9001, ISO 14001, SA 8000, ISO 26000, and GRI related studies should be included in the literature. Please align your study objectives with the updated literature contribution.

Author Response 3.1

Thank you for this continued feedback. We have revised the manuscript to explain how our study's focus, which is on government-led supply chain regulations in middle-income countries, is connected to prior scholarly work in the realm of private-sector sustainability standards such as the GRI or ISO certifications.

Reviewer #4

Reviewer Comment 4.1

Thank you. My concerns are addressed by the authors in the revised manuscript.

Author Response 4.1

Thank you again for your constructive feedback on this manuscript.

REVIEWERS' COMMENTS

Reviewer #2 (Remarks to the Author):

Thank you for the revision. I have no further comments.